# Efficient *in vivo* targeting of the myocardial scar using Moloney murine leukaemia virus complexed with nanoparticles

Timo Mohr[1,2], Miriam Schiffer[1,2], Deepak Ramanujam[3,4] ⓘD, Esther Carls[1],
Callum M. Zgierski-Johnston[5] ⓘD, Thomas Kok[5] ⓘD, Pia Niemann[2], Caroline Geisen[2], Peter Kohl[5] ⓘD,
Stefan Engelhardt[3,4], Bernd K. Fleischmann[2] ⓘD and Wilhelm Roell[1] ⓘD

[1]*Department of Cardiac Surgery, University Hospital Bonn, University of Bonn, Bonn, Germany*
[2]*Institute of Physiology I, Medical Faculty, University of Bonn, Bonn, Germany*
[3]*Institute of Pharmacology and Toxicology, Technical University of Munich (TUM), Munich, Germany*
[4]*DZHK (German Centre for Cardiovascular Research), partner site Munich Heart Alliance, Munich, Germany*
[5]*Institute for Experimental Cardiovascular Medicine, University Heart Center Freiburg -Bad Krozingen, Medical Faculty and Medical Center, University of Freiburg, Freiburg im Breisgau, Germany*

Handling Editors: Bjorn Knollmann & T Alexander Quinn

The peer review history is available in the Supporting Information section of this article (https://doi.org/10.1113/JP288020#support-information-section).

T. Mohr and M. Schiffer contributed equally to this work.

Ethical statement: All animal experiments were performed according to National Institutes of Health animal protection guidelines and were approved by local authorities (Landesamt für Natur und Verbraucherschutz Nordrhein-Westfalen, NRW; animal protocol numbers: 81.02.04.2018.A120 and 81.02.04.2023.A270; Regierungspräsidium Freiburg, animal protocol number: G22/110).

**Abstract figure legend** The complexation of Moloney murine leukaemia virus (MMLV) with magnetic nanoparticles (MNP) greatly enhanced the transduction of (myo)fibroblasts (FB), when applied in a magnetic field, both *in vitro* and *in vivo* in cryoinjured mice. Targeting of myocardial scar FB *in vivo* with a MMLV-connexin 43 (Cx43) construct demonstrated an increased expression of Cx43 and protection against the induction of ventricular tachycardia at 2 and 8 weeks post-transduction. Created with BioRender.com and Smart.servier.com.

**Abstract** During myocardial infarction, native myocardium is replaced by a fibrous scar, impairing cardiac pump function and leading to potentially life-threatening ventricular tachycardias. Gene therapy-based targeting of the cardiac scar is essential for short- and long-term treatment of post-infarct sequelae. However, there are currently no effective methods to target and transduce cardiac (myo)fibroblasts (FB) *in vivo*. Therefore, Moloney murine leukaemia virus (MMLV) encoding for the fluorescent reporter mCherry complexed with magnetic nanoparticles (MNP) in combination with magnetic steering was tested. This approach strongly increased the transduction rate of FB by four-fold *in vitro*. Additionally, injection of MMLV/MNP complexes into the forming scar during exposure of the heart to a magnetic field to increase virus dwell-time 3 days after left ventricular cryo-injury, a time when FB proliferation in the infarct peaks, yielded efficient transduction of resident FB. We further assessed the functional impact of overexpressing the gap junction protein connexin 43 (Cx43). MMLV-mediated Cx43 overexpression (MMLV-Cx43) in FB increased the formation of functional gap junctions *in vitro* and substantially lowered post-cryo-injury ventricular tachycardia incidence (by 50%), as demonstrated by *in vivo* electrophysiological testing 2 and 8 weeks after MMLV/MNP injection into the lesion. This anti-arrhythmic effect was probably the result of a decrease in the heterogeneity of conduction around and across the scar, as observed by optical mapping in isolated hearts overexpressing Cx43. Thus, MMLV/MNP complexes combined with magnetic steering offer an efficient strategy for targeting and transducing FB *in vitro* and *in vivo*, allowing modulation of the functional properties of cardiac scars.

(Received 31 October 2024; accepted after revision 11 June 2025; first published online 21 July 2025)

**Corresponding author** W. Roell: Department of Cardiac Surgery, University Hospital Bonn, Venusberg-Campus 1, 53127 Bonn, Germany. Email: wroell@uni-bonn.de B. K. Fleischmann, Institute of Physiology I, Medical Faculty, University of Bonn, Venusberg-Campus 1, 53127 Bonn, Germany. Email: bernd.fleischmann@uni-bonn.de

**Key points**

- Genetic targeting and efficient transduction of (myo)fibroblasts (FB) *in vivo* are necessary to modify the properties of cardiac scars for therapeutic benefit. However, this has proven highly inadequate because of a lack of suitable viral vectors.
- We complexed Moloney murine leukaemia virus (MMLV) with magnetic nanoparticles (MNP) and applied a magnetic field to achieve prominent *in vitro* transduction of FB.
- Magnet-assisted *in vivo* injections of MMLV/MNP complexes into the developing scar enabled efficient transduction of cardiac tissue-resident FB.
- MMLV-Cx43-based overexpression of connexin 43 increased the density of functional gap junctions in FB *in vitro*. Following direct injection of the virus into the developing cardiac scar in a murine cryo-lesion model, electrophysiological *in vivo* testing showed that the incidence of ventricular tachycardia was reduced by 50% at 2 and 8 weeks post-treatment with MMLV.
- Our approach enables efficient targeting and transduction of FB in the cardiac scar, providing a blueprint for translation.

## Introduction

Myocardial ischaemia irreversibly damages the myocardium within a few hours, resulting in myocardial infarction (MI), one of the leading causes of death worldwide (Khan et al., 2020). During the scarring process that follows, dead cardiomyocytes (CM) are replaced by cardiac (myo)fibroblasts (FB), which are the dominant

cell type within the scar area (Frangogiannis, 2014; Fu et al., 2018; Ivey & Tallquist, 2016). Because FB are non-contractile, cardiac pump function deteriorates in the presence of large scars, leading to heart failure, which has a poor prognosis (Cao et al., 2024). Besides heart transplantation, there is no causal cure for heart failure because the available pharmacological treatments only alleviate symptoms, and the long-term benefit of assist devices is still limited (McDonagh et al., 2023; McNamara et al., 2021). Furthermore, although cardiac FB express gap junction proteins (Camelliti, Devlin et al., 2004; Camelliti, Green et al., 2004) and couple to cardiac myocytes in cardiac lesions (Quinn et al., 2016; Rubart et al., 2018; Schultz et al., 2019), they are not excitable (no fast voltage-dependent $Na^+$ channels), resulting in electrical heterogeneity and isolation of the infarcted area (Agullo-Pascual et al., 2014; Kohl, 2003). This can cause life-threatening ventricular tachycardias (VT) and fibrillation (VF), both in the acute phase and in the long term after MI (Jansen et al., 2012; Nisbet et al., 2016; van Rijen et al., 2004). Implanted defibrillators can detect and treat ventricular arrhythmias (Moss et al., 2002). However, patients still have a reduced quality of life and life expectancy, depending on the frequency and severity of VT (Sesselberg et al., 2007).

So far, therapeutic approaches for treating post-MI sequelae have focused on CM. Progress has been made in recent years targeting the myocardium, taking advantage of adeno-associated viruses (AAV), which efficiently transduce CM in small and large animals. However, the systemic administration of AAV also transduces skeletal muscle and liver cells, potentially leading to serious side effects (Ertl, 2022). Moreover, AAV-based gene therapies in patients suffering from severe heart failure did not achieve the anticipated success (Hulot et al., 2016). Because the clinical end-point was not met, trials were stopped with a proposed reason of insufficient transduction of CM by AAV (Lyon et al., 2020).

Instead of targeting CM, more recently, the scientific focus has shifted to non-myocyte-related heart disease mechanisms in general and, for cardiac scars, to the potential impact of modulating the molecular and/or functional characteristics of cardiac FB in particular.

However, unlike the experimental success achieved in targeting CM *in vivo* using AAV, the transduction of cardiac FB, especially in cardiac scars, has proven ineffective. In earlier work, we targeted the scar by injecting lentivirus directly into the developing lesion in mice, which resulted in the transduction of only very low numbers of FB (1.1–2.8%) (Roell et al., 2018). Therefore, we tested Moloney murine leukaemia virus (MMLV) to improve large-scale transduction of cardiac FB because this virus has been reported to transduce preferentially cardiac FB and endothelial cells (EC) but not CM after intrapericardial injection in newborn mice (Ramanujam et al., 2016).

Here, we explore the utility of MMLV for transducing cardiac FB in adult mice *in vivo* using two constructs: one encoding the expression of mCherry (MMLV-mCherry) and another that encodes both mCherry and connexin 43 (Cx43), connected via a P2A element (MMLV-Cx43) (Niemann et al., 2022). Building on our earlier work, we examined the impact of complexing MMLV with magnetic nanoparticles (MNP) to further enhance transduction efficiency. We show that combining MMLV/MNP complexes with a strong and focused magnetic field can increase transduction efficiency four-fold, compared to overnight incubation with MMLV alone, leading to the formation of functional gap junctions in FB *in vitro* when using MMLV-Cx43. The injection of MMLV/MNP complexes into the forming scar *in vivo*, combined with magnetic steering, yielded efficient transduction of resident scar FB and strong anti-VT protection upon injection of MMLV-Cx43.

## Methods

### Ethical approval

All animal experiments were performed in accordance with the ARRIVE guidelines, the guidelines of the German animal protection act, and were approved by the animal welfare committee of the University of Bonn and local government authorities (Landesamt für Natur und Verbraucherschutz Nordrhein-Westfalen, NRW; animal protocol numbers: 81.02.04.2018.A120

**Timo Mohr** obtained his MSc in biology from the Technical University of Munich specializing in medical biology. He is currently pursuing his PhD in experimental medicine at the Department of Cardiac Surgery, UKB, in close collaboration with the Institute of Physiology I, University of Bonn. He is focusing on gene therapeutic approaches to target cardiac fibroblasts *in vitro* and *in vivo* with to modify the functional, specifically the electrophysiological properties of infarcted heart. **Miriam Schiffer** is a postdoctoral researcher at the Department of Cardiac Surgery, at the UKB and the Institute of Physiology I, at the University of Bonn. She completed her PhD in Experimental Medicine at the University of Bonn in 2023. In her PhD project entitled 'Viral Gene Transfer of Ischemic Myocardium using Magnetic Nanoparticles', she injected *ex vivo* lentivirally transduced connexin43-overexpressing cardiac fibroblasts into the lesioned mouse heart. Currently, she is characterizing newly established connexin43 overexpressing iPS-cells.

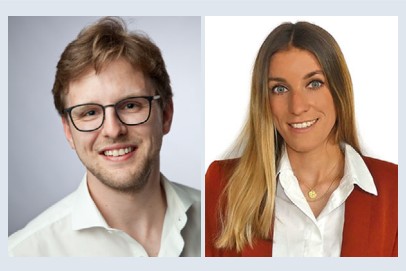

and 81.02.04.2023.A270; Regierungspräsidium Freiburg, animal protocol number: G22/110). Before the authorities approved the animal protocols, it was first determined whether the '3 R rules-Replace, Reduce, Refine' were integrated in the experimental design. Mice were kept prior to and following surgery in groups of up to five individuals, with access to food and water *ad libitum* in a temperature- and humidity-controlled facility under a 12:12 h light/dark photocycle. The authors understand the ethical principles of *The Journal of Physiology* and the work presented complies with the Journal's policies regarding animal experiments.

### Cell culture

All cells were cultured at 37°C and in 5% $CO_2$.

NIH/3T3 mouse FB (3T3; ATCC, Manassas, VA, USA; #CRL-1658) were cultured in Dulbecco's modified Eagle's medium (DMEM; Gibco, Karlsruhe, Germany; #41965039), supplemented with 10% v/v foetal calf serum (FCS; Gibco; #16000044), 100 U $mL^{-1}$ penicillin, 100 µg $mL^{-1}$ streptomycin (Thermo Fisher Scientific, Waltham, MA, USA; #15140-122), 0.1 mM non-essential amino acids (Thermo Fisher Scientific; #11140-035) and 0.1 mM $\beta$-mercaptoethanol ($\beta$-ME; Sigma-Aldrich, St Louis, MO, USA; #M3148) in tissue culture flasks.

C166 mouse EC (C166; ATCC; #CRL-2581) were maintained in DMEM, supplemented with 10% v/v FCS, 100 U $mL^{-1}$ penicillin, 100 µg $mL^{-1}$ streptomycin and 1 mM sodium pyruvate (Gibco; #11360039).

Primary embryonic cardiac FB (eFB) were isolated from CD1 embryonic mouse hearts on embryonic day (E)12.5 to E13.5 (breeding pairs were supplied by Charles River Laboratories, Sulzfeld, Germany). After dissection, hearts were minced using spring scissors and enzymatically digested with collagenase type II (520 U $mL^{-1}$; Worthington Biochemical Corporation, Lakewood, NJ, USA; #NC9693955) for 45 min on a shaker at 37°C and 400 rpm. Then, $1.0 \times 10^7$ cells were seeded onto a 0.1% gelatine (Sigma-Aldrich; #G2500) coated T75 cell culture flask and cultured in 10% DMEM (as 3T3). After 3 days, cells were split 1:2 in freshly coated T75 cell culture flasks using Accutase (Millipore, Burlington, MA, USA; SCR005) at room temperature (RT) for 5 min. After three additional days, eFB were seeded at a concentration of $2.0 \times 10^4$ (in 24-well plates for histological analyses) or $2.0 \times 10^5$ [in six-well plates for western blotting (WB)] for transduction on the following day.

### Generation of MMLV-Cx43-P2A-mCherry and MMLV-mCherry viruses

For the generation of the MMLV-Cx43 vector, the Cx43-P2A-mCherry-pMK-RQ plasmid previously

described (Niemann et al., 2022) was used. The vector contains the murine Cx43 cDNA, linked in frame to the mCherry cDNA via the DNA sequence of the self-cleaving GSG-P2A peptide. The insert was cut out and cloned into the multiple cloning site of the pMXs retroviral expression vector (Kitamura et al., 2003). For generation of the MMLV-mCherry vector, the mCherry fragment was amplified from the insert by polymerase chain reaction and was cloned into the pMXs vector. Preparation, purification, and titre determination of MMLV were performed as previously described (Ramanujam et al., 2016).

### Virus/MNP complexation

Silica-iron oxide magnetic nanoparticles (SoMag5-MNP) consisting of a magnetite core and a silica iron oxide shell (inorganic core diameter $7 \pm 1$ nm, hydrodynamic diameter $40 \pm 14$ nm, $\zeta$-potential $-38$ mV, provided by Dr. C. Plank, Technische Universität München) were complexed with MMLV in Hanks Balanced salt solution (HBSS$^{++}$; Gibco; #14025050) for 20 min at RT. For *in vitro* transduction, complexes were prepared in a maximum volume of 200 µL [500 virus genomes (VG)/cell + 500 fg Fe SoMag5/VG, 250VG/cell + 500 fg Fe SoMag5/VG or 250 VG/cell + 250 fg Fe SoMag5/VG] and used in a volume of 10 µL (24-well) or 50 µL (six-well) per well. For *in vivo* transduction, complexes were prepared in a maximum volume of 30 µL (250 VG + 250 fg Fe/VG, calculated for $2.5 \times 10^5$ cells/lesion) and 5 µL were injected per mouse.

### *In vitro* transduction of cells

For *in vitro* transduction, $2.0 \times 10^4$ cells (3T3, eFB or C166) were seeded per well onto glass coverslips coated with 0.1% gelatine in 24-well plates and cultured for 24 h in the respective media.

For overnight transduction with MMLV alone, virus particles, diluted in HBSS$^{++}$, were directly added to the medium in the wells. Culture plates were incubated for 30 min on a shaker (50 rpm), and afterwards maintaind for 20 h without shaking.

MMLV (250 or 500 VG per cell) + SoMag5-MNP (250 or 500 fg/VG) complexes were prepared as described above just before transduction. Cells were washed with phosphate buffered saline (PBS; Gibco; #11503387), covered with HBSS$^{++}$ (30 µL for 24-well plates or 300 µL for six-well plates, per well), and MMLV/MNP complexes were added. For transduction using magnet steering, culture dishes were placed on a magnetic plate (MagnetoFACTOR-24; Chemicell, Berlin, Germany; #9009) and incubated on a shaker (50 rpm) for 10, 30, 45

or 60 min. Without magnetic steering, culture plates were incubated on a shaker (50 rpm) for 30 min.

At the end of each incubation, the supernatant was removed and replaced by fresh medium (DMEM, 10% FCS).

### Intramyocardial injection and magnetic steering of MMLV/SoMag5-MNP complexes

Before the intramyocardial injection of MMLV/SoMag5-MNP complexes, left ventricular cryoinjuries (CI) were generated in CD1 mice aged 10–12 weeks (Charles River Laboratories). Because transmural myocardial lesions were essential for the comparability of subsequent functional and electro-physiological analyses of the different study groups, the CI-model was chosen. For this purpose, mice were intubated under general inhalation anaesthesia (1–2 vol% isoflurane (IsoFlo; Zoetis Inc., Kalamazoo, Michigan, USA; #50019100) and 0.8 L min$^{-1}$ oxygen), positioned on a heating pad, where a left lateral thoracotomy was carried out. A round copper probe (diameter 3.5 mm, precooled in liquid $N_2$) was pressed onto the dry-blotted antero-lateral left ventricular wall three times for 15 s each. Subsequently, the chest was closed in layers and de-aired using a chest drain to prevent pneumothorax. After extubation, mice were placed under a heating lamp to support further recovery from the operative procedure.

Three days after CI, again under general anaesthesia, the initial thoracic incision was re-opened as described before. MMLV-complexes were injected into the centre of the lesion using a 34 G injection needle. During and for 10 min after the injection, a bar magnet (1.3 T) was placed above the injection site at a distance of 5 mm. Subsequently, the chest was closed as described above. For both surgeries, analgesia (0.1 mg kg$^{-1}$ I.P., Buprenovet sine 0.3 mg mL$^{-1}$; Richter Pharma AG, Wels, Austria) was administered 30 min preoperatively and three times daily up to the third postoperative day. To prevent a virus-mediated prominent immune response, mice received 1 mg kg$^{-1}$ prednisolone I.P. (Prednisolut 100 mg; Mibe, Brehna, Germany; #14142100, diluted in 5% glucose solution (Glucosteril 5%; Fresenius Kabi, Bad Homburg, Germany; #00424585) perioperatively and once a day for 10 days postoperatively. The timeline of the *in vivo* transduction experiments is depicted in Scheme 1.

The 1.3 T rod magnet used in this study was already utilized in a previous study (Ottersbach et al., 2018). In that study, it had been shown that the magnetic field generated targets a tissue area of 3.14 mm$^2$ (equal to 1.60 mm$^3$ at an average scar thickness of 0.51 mm). Overall nucleus count (irrespective of cell identity) within the scar 17 days after CI was determined in two CI hearts (five sections a 10 μm each; apical, apicomedial, medial, mediobasal and basal) and amounted to ∼3200 cells per mm$^2$ of the 10 μm thick tissue section analysed, or 320 000 cells mm$^{-3}$ scar volume. Therefore, the scar volume affected by the magnetic field contained ∼500,000 (1.6 × 320,000) cells, mostly FB. To transduce ∼50% of the FB (250,000 cells) with MMLV/MNP complexes with the optimal concentration of 250 VG/cell + 250 fg Fe/VG, as determined by *in vitro* experiments, a total amount of 6.25 × 10$^7$ VG (250,000 cells × 250 VG MMLV) and 15.62 μg of Fe (6.25 × 10$^7$ VG × 250 fg Fe SoMag5), complexed in 5 μL of HBSS$^{++}$, were injected intramyocardially and enriched locally using the magnetic field described.

### *In vivo* echocardiography and electrophysiology

Transthoracic echocardiography was performed 16 or 58 days after initial CI (i.e. 13 or 55 days after MMLV injection) under inhalation anaesthesia using a Philips CX50 ultrasound system (ATL-Philips, Oceanside, CA, USA) equipped with a 15 MHz transducer. For assessment of left ventricular function, short-axis M-mode was recorded at the level of the papillary muscles and three still images were analysed for anterior wall thickening and fractional shortening.

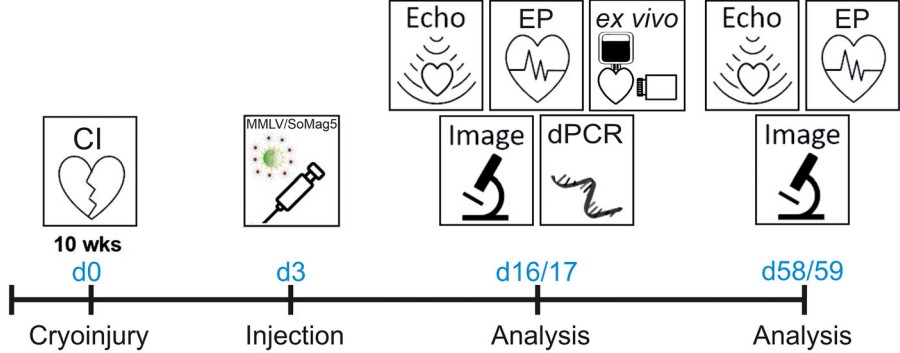

**Scheme 1. Timeline of the *in vivo* transduction experiments**
For details, see text.

*In vivo* electrophysiology testing was performed under inhalation anaesthesia 17 or 59 days after initial CI (i.e. 14 or 56 days following MMLV injection), as reported earlier (Roell et al., 2007, 2018). In addition to the recording of four-lead surface ECG (PowerLab 16/30, LabChart 7; ADInstruments, Pty Ltd, Bella Vista NSW, Australia), the tip of a 2 Fr octapolar mouse-electrophysiological catheter (CIBER Mouse Electrophysiology Catheter; NuMED, Hopkinton, NY, USA) was inserted via the right jugular vein to the apex of the right ventricle and bipolar intra-cardiac electrograms were recorded from neighbouring electrode pairs simultaneously at the atrial, His-bundle and ventricular levels. Trains of rectangular stimulus pulses were applied via the apical electrode pair, using a multi programmable stimulator (Model 2100; A-M Systems, Carlsborg, WA, USA; Stimulus 3.4 Software; Institute for Physiology I, University of Bonn, Bonn, Germany). Extra-stimuli and burst-stimuli protocols were used as follows: extra-stimulus pacing was applied to trigger up to three extra beats (5 ms stepwise S1–S2 or S2–S3 reduction, starting at 90 ms interstimulus duration) at S1–S1 cycle lengths of 120, 110 and 100 ms. Burst-stimulus pacing was performed for $3 \times 0.5$ s and $3 \times 1$ s (S1–S1 cycle lengths starting at 50 ms; stepwise reduction by $10 \times 10$ ms). Between stimulation runs, mice were allowed to recover in sinus rhythm (RR interval under inhalation anaesthesia ∼120–150 ms) for at least 4 s. For analysis, according to the clinical definition and previous work (Roell et al., 2007, 2018) VT was characterized by at least four consecutive ventricular extra-beats with atrioventricular dissociation or episodes of VT/VF.

Mice were killed (cervical dislocation under inhalation anaesthesia) after measurements and hearts were swiftly excised for further analyses.

### Histological staining and microscopy

3T3, eFB or C166 cells were seeded ($2.0 \times 10^4$ cells per well) onto glass coverslips (0.1% gelatine-coated) in 24-well plates and cultured for 24 h in their respective media before transduction. Transduced cells were cultured for an additional 3 days and then fixed with 4% formaldehyde (FA; Sigma-Aldrich; #1.00496) for 20 min at RT.

Excised hearts were macroscopically imaged (Axio Zoom V16; Zeiss, Jena, Germany; software: ZEN3.1, blue edition; Zeiss), subsequently fixed by submersion in 4% FA for 1–2 h at 4°C, incubated with 20% Sucrose (Sigma Aldrich; #S0389) overnight, embedded in Tissue-Tek OCT compound (Sakura Finetek Germany GmbH, Staufen, Germany; #4583) and frozen in dry-ice cooled isopentane. Frozen hearts were cut into slices (10 μm) using a cryostat (CM3050S; Leica, Wetzlar, Germany).

Prior to immunostaining, cells and heart slices were permeabilized and blocked with 0.2% Triton X (Sigma-Aldrich; #93418) and 5% donkey serum (Jackson ImmunoResearch, West Grove, PA, USA; #017-000-121) in PBS (PBS tablets; Gibco; #11510546) for 20 min at RT. Primary antibodies (Table 1) were diluted in 5% donkey serum in PBS and tissue was incubated overnight at 4°C. After washing (PBS), secondary antibodies (Table 2) in PBS with 1 μg mL$^{-1}$ Hoechst (Sigma-Aldrich; #B2261) were incubated for 1 h at RT. After washing, cells and slices were mounted with Aqua-Polymount (Polysciences, Warrington, PA, USA; #18606). Fluorescence images (transduced cells and overview of slices) were taken using a Zeiss Axiovert 200 microscopy system (ZEN3.1, blue edition; Zeiss) or a Keyence BZ-X810 fluorescence microscope (Keyence, Osaka, Japan; software: BZ-X800 Analyser; Keyence). High resolution fluorescence images were taken using a confocal microscope (Nikon, Tokyo, Japan; Eclipse Ti and A1R MP system; software: NIS-Elements AR 5.11.01; Nikon).

For visualization of internalized iron, Prussian blue in combination with Eosin staining (detection of acidophilic cell structures) was performed. In brief, 5% hydro-chloric acid and 5% potassium hexacyanoferrate (II) were mixed (1:1) and fixed heart sections were incubated for 20 min, washed with H$_2$O bidest and mounted with resin based medium before brightfield images were taken (Axio Zoom V16; Zeiss, software: ZEN3.1, blue edition; Zeiss).

### Analysis of scar size

Scar size was determined by measuring the epicardial scar surface on macroscopic images of the heart ($8\times$; Axio Zoom V16; Zeiss) using ImageJ (NIH, Bethesda, MD, USA). For morphometric analysis of scar volume, cryosections of the hearts were prepared from apex to base as described above and stained with Sirius Red (Sigma-Aldrich; #365548) and Fast-Green (Thermo Fisher Scientific; # F-9910). In brief, sections were fixed in pre-warmed Bouin's solution (Sigma-Aldrich; #HT10132) for 1 h at 55°C. After washing, slides were incubated in Fast Green (0.1% in H$_2$O) for 10 min and shortly destained in 1% acetic acid at RT. Subsequently, sections were stained with Sirius Red (0.1% in picric acid) for 30 min and mounted with resin-based medium. The area of vital left ventricular myocardium and collagen-rich scar area was measured every 400 μm in these sections using ImageJ software ($3.2\times$; Axio Zoom V16; Zeiss) to extrapolate the scar volume. The scar volume is given as a percentage of total left ventricular volume (inclusive of the intra-ventricular septum).

**Table 1. Primary antibodies for histological stainings (IHC) and WB analysis.**

| Antigen (clone) | Isotype | Application/dilution | Catalogue number | Manufacturer | Reference |
|---|---|---|---|---|---|
| CD45 (IBL-5/25) | Rat monoclonal IgG | IHC: 1:400 | 05–1416 | Millipore, Burlington, MA, USA | Garrison et al. (2024) |
| Cx43 | Rabbit polyclonal antibody; immunizing peptide sequence: CDQRPSSRASSRA-SSRPRPDDLEI | IHC: 1:800 WB: 1:3000 | custom-produced | PSL GmbH, Heidelberg, Germany | Niemann et al. (2022); Roell et al. (2018) |
| GAPDH (GA1R) | Mouse monoclonal IgG1 | WB: 1:3000 | MA5-15738-A555 | Thermo Fisher Scientific, Waltham, MA, USA | Dey et al. (2020) |
| mCherry | Rabbit polyclonal | IHC: 1:500 | NBP2-25157 | Novus Biologicals, Littleton, CO, USA | Miltner et al. (2019) |
| mCherry | Chicken polyclonal IgY | IHC: 1:500 | NBP2-25158 | Novus Biologicals | Zhou et al. (2018) |
| PDGFRα | Goat polyclonal IgG | IHC: 1:200 | AF1062 | R&D Systems, Minneapolis, MN, USA | Waisman et al. (2021) |
| P2A (3H4) | Mouse monoclonal IgG1 kappa | IHC: 1:700 | NBP2-59627 | Novus Biologicals | Niemann et al. (2022) |
| PECAM (MEC13.3) | Rat monoclonal IgG2a kappa | IHC 1:400 | 550274 | BD Biosciences, Franklin Lakes, NJ, USA | Petrillo et al. (2023) |
| Tcf21 | Rabbit polyclonal IgG | IHC: 1: 200 | NBP1-88637 | Novus Biologicals | Coppiello et al. (2023) |

## Quantification of transduction efficiency and cell specificity

Cells were fixed on glass coverslips, as described above and incubated with 1 µg mL$^{-1}$ Hoechst solution (Sigma-Aldrich) for 10 min at RT. Fluorescence images were taken (20×; Keyence BZ-X810; Keyence). The number of mCherry$^+$ cells and nuclei were counted in a defined area (3T3-FB, C166: 5.5 mm$^2$, eFB: 10.5 mm$^2$). Transduction efficiency was calculated as a ratio of mCherry$^+$ cells per nuclei count in the area.

For all quantifications of MMLV transductions *in vivo*, heart sections were stained for mCherry.

To quantify the number of MMLV-transduced cells within the scar, nuclei of mCherry$^+$ cells were counted in 20–32 10 µm sections (with a distance of ∼100 µm between sections) from MMLV-mCherry and MMLV-Cx43 hearts ($n = 3$ each) covering the entire area containing transduced cells. To determine the total number of transduced cells within the lesion area, nuclei of mCherry$^+$ cells, which were counted in the 20–32 representative sections, distributed over the entire lesion. Calculation of total mCherry$^+$ cells within the entire transduction area was based on section thickness and distance in between representative sections.

Local MMLV transduction *in vivo* was determined in the same hearts (five sections per heart, covering the transduced area) by counting mCherry$^+$ cells and total number of nuclei in the transduced area (excluding peripheral scar area), and calculated as the ratio of mCherry$^+$ cells relative to the total number of nuclei.

For cell type determination of mCherry$^+$ cells *in vivo*, cardiac sections were stained for several markers: (1) PECAM, mCherry; (2) CD45, mCherry; (3) PDGFRα, αSMA, mCherry; (4) PDGFRα, Tcf21, mCherry; and (5) α-actinin, mCherry. Confocal images were taken and mCherry$^+$/marker$^+$ cells (single or double positive for marker) were counted.

To assess FB markers, MMLV-mCherry transduced hearts ($n = 3$, 1 section each) were stained for the marker combinations 3 and 4. The proportion of marker$^+$ cells in the scar was calculated as the ratio of the number of mCherry$^+$/marker$^+$ cells per total number of mCherry$^+$ cells.

To assess EC transduction in the scar, MMLV-mCherry transduced hearts ($n = 3$, 3 sections each) were stained for

**Table 2. Secondary antibodies histological stainings and WB analysis.**

| Isotype and fluorochrome | Species | Application/dilution | Catalogue number | Manufacturer | Reference |
|---|---|---|---|---|---|
| Anti-rabbit IgG (H+L) Alexa Fluor 488/Cy2 conjugated | donkey | IHC: 1:400 WB: 1:300 | 711–545-152 | Jackson ImmunoResearch, West Grove, PA, USA | IHC: D'Amour et al. (2020) WB: Sanchez-Arias et al. (2019) |
| Anti-chicken IgY (IgG) (H+L) Cy2 conjugated | donkey | IHC: 1:400 | 703–225-155 | Jackson ImmunoResearch | Santos-Ferreira et al. (2016) |
| Anti-goat IgG (H+L) Cy2 conjugated | donkey | IHC: 1:400 | 705–225-147 | Jackson ImmunoResearch | Ryan et al. (2017) |
| Anti-chicken IgY (IgG) (H+L) Cy3 conjugated | donkey | IHC: 1:400 | 703–165-155 | Jackson ImmunoResearch | Duval et al. (2018) |
| Anti-rabbit IgG (H+L) Cy3 conjugated | donkey | IHC 1:400 | 711–165-152 | Jackson ImmunoResearch | Exposito-Alonso et al. (2020) |
| Anti-mouse IgG (H+L) Alexa Fluor 647 conjugated | donkey | IHC: 1:400 | 715–605-151 | Jackson ImmunoResearch | Klein et al. (2020) |
| Anti-mouse IgG Fc$\gamma$ subclass 1 Alexa Fluor 647 conjugated | goat | IHC 1:400 | 115–605-205 | Jackson ImmunoResearch | Waisman et al. (2021) |
| Anti-mouse IgG2a (H+L) Alexa Fluor 647 conjugated | goat | IHC: 1:400 | 115–605-206 | Jackson ImmunoResearch | Sluka et al. (2020) |
| Anti-goat IgG (H+L) Alexa Fluor 647 conjugated | donkey | IHC: 1:400 | 705–605-147 | Jackson ImmunoResearch | Marques et al. (2020) |
| Anti-rat IgG (H+L) Alexa Fluor 647 conjugated | donkey | IHC: 1:400 | 712–605-153 | Jackson ImmunoResearch | Diéguez-Hurtado et al. (2019) |

PECAM and mCherry. The proportion of transduced EC was calculated as the ratio of PECAM$^+$/mCherry$^+$ cells per total number of mCherry$^+$ cells.

### Digital PCR

For transgene expression analysis on the mRNA level, digital PCR (dPCR) experiments were performed (QIAcuity One System; Qiagen, Hilden, Germany) in accordance with the manufacturer's instructions. For that purpose, scars were excised 14 days post viral injection under a stereomicroscope (Wild Heerbrugg Leica M8 Stereozoom; Leica), incubated in RNAlater solution (Sigma-Aldrich; #R0901) overnight at 4°C, and subsequently frozen in liquid N$_2$. RNA was isolated using the RNAeasy Plus Micro Kit in accordance with the manufacturer's instructions (Qiagen; #74034). To evaluate RNA quality, the RNA integrity number was determined by a Bioanalyzer 2100 (Agilent Technologies, Santa Clara, CA, USA) using the Agilent RNA 6000 Nano Kit (#5067-1511) and only samples with a RNA integrity number >7.5 were analysed further. RNA

was quantified with the NanoQuant Plate (Tecan Spark microplate reader; Tecan, Männedorf, Switzerland) and 50 ng RNA per sample was transcribed into cDNA (20 μL reaction volume, SuperScript VILO cDNA Synthesis Kit; Invitrogen, Thermo Fisher Scientific; #11754050). dPCR was performed in QIAcuity 24-well 8.5k Nanoplates (Qiagen; #250011) with 1.5 μL of cDNA, PCR Master-Mix (QIAcuity Probe PCR Kit; Qiagen; #250103) a VIC-labelled Hprt housekeeper TaqMan assay (Thermo Fisher Scientific; #4448 490) and a FAM-labelled custom-made exogenous Cx43 TaqMan assay, specifically detecting a cDNA sequence within 210 bp from the Cx43 3'-end to the mCherry 5'-end (Thermo Fisher Scientific). Partitioning and imaging (exposure time: 500 ms, gain: 6) were performed automatically in the QIAcuity One instrument for end-point PCR after 40 cycles: dPCR: (I) 95°C for 2 min, 40 cycles of (II–III): (II) 95°C for 15 s, (III) 60°C for 30 s). The copy number of the target cDNA was normalized to the housekeeping gene Hprt as:

$$\text{Relative copy number} = \frac{\text{Concentration (target)} \left(\frac{\text{copy}}{\mu l}\right)}{\text{Concentration (Hprt)} \left(\frac{\text{copy}}{\mu l}\right)}$$

## Western blot analysis of mCherry, Cx43 and P2A protein expression

For WB analysis, 3T3 and eFB were seeded ($2.0 \times 10^5$ cells per well) in six-well plates (0.1% gelatine-coated for eFB) and MMLV/MNP transduction was performed as described above. *In vitro* samples were lysed on ice in freshly prepared RIPA buffer (50 mM Tris HCl (pH 7.5) (Carl Roth, Karlsruhe, Germany; #9090.3); 1% IGEPAL CA-630 (Sigma-Aldrich; #I8896); 0.25% sodium deoxycholate (Sigma-Aldrich; #D6750-100); 150 mM NaCl (Carl Roth; #3957.1); 1 mM EDTA (Applichem, Darmstadt, Germany; #A4892,0100); 1 mM phenylmethanesulfonyl fluoride (PMSF; Sigma-Aldrich; #P7626); protease and phosphatase inhibitors (Thermo Fisher Scientific; #A32961).

Protein concentrations were determined using Pierce BCA Protein assay kit (Thermo Fisher Scientific; #23225). Sodium dodecyl sulfate (SDS) minigels (12% separating gel:12% acrylamide (Bio-Rad, Feldkirchen, Germany; #1610154), 375 mM Tris HCl (pH 8.8), 0.1% SDS (Carl Roth; #0183.2), 0.05% APS (Bio-Rad; #1610700), 0.05% TEMED (Sigma-Aldrich; #T7024); 8% stacking gel: 8% acrylamide in 125 mM Tris HCl, pH 6.8, 0.1% SDS, 0.05% APS, 0.1% TEMED) were used. Protein lysates were mixed with Laemmli Buffer ($2\times$ or $4\times$; Bio-Rad; #1610-737 or #161-0747; supplemented with $\beta$-ME (Sigma-Aldrich; #M3148) and denatured for 10 min at 99°C. Prepared lysates and protein markers (Precision Plus Protein WesternC Standards, Bio-Rad; #1610376) were separated by gel electrophoresis (ProSieve EX Running buffer $10\times$; Lonza, Basel, Switzerland; #00200307, diluted in $H_2O$ bidest). Separated proteins were blotted (ProSieve EX Transfer buffer $10\times$; Lonza; #002 00309, diluted in $H_2O$ bidest) onto a methanol-activated low-fluorescence 0.2 μm PVDF-membrane (Advansta, San Jose, CA, USA; #L-08001-10) (75 min, 100 V) using a tank blot (ProSieve EX Transfer buffer $10\times$; Lonza; #002 00309, diluted in $H_2O$ bidest). Membranes were washed once in Tris-buffered saline (TBS) [20 mM Trizma Base; Sigma-Aldrich; #T1503], 150 mM NaCl (Carl Roth; #3957.1) and blocked in 5% skim-milk-powder (VWR, Radnor, PA, USA; #84 615.0500) in TBS-T [0.1% Tween-20 (Sigma-Aldrich; #P9416) in TBS] for a minimum of 1 h. Membranes were incubated overnight at 4°C with primary antibodies (Table 1), diluted in 5% MP/TBS-T. After washing in TBS-T, membranes were incubated for 1 h at room temperature with secondary antibodies (Table 2) as well as Precision Protein StrepTactin-HRP Conjugate (dilution 1:3000; Bio-Rad; #1610381), diluted in 5% MP/TBS-T. After washing in TBS-T, signals were detected with the ChemiDoc MP Imaging System (Bio-Rad). Blots with peroxidase-conjugated secondary antibodies were incubated with Pierce-ECL Western Blotting Substrate (Thermo Fisher Scientific; #32106) prior to detection. Semi-quantitative analysis of target protein expression was performed with ImageLab, version 5.2.1 (Bio-Rad) and was normalized to GAPDH (cells).

## Fluorescence recovery after photobleaching

For fluorescence recovery after photobleaching (FRAP) experiments, 3T3 were seeded ($2.0 \times 10^5$ cells per well) in six-well plates and MMLV/MNP (MMLV-Cx43 or MMLV-mCherry) transduction was performed as described above. Three days post transduction, 3T3 were incubated for 20 min at 37°C with Calcein-AM (0.38 μM; Invitrogen, Thermo Fisher Scientific; #C3100MP) diluted in 10% DMEM. The fluorescence intensity of clusters of Calcein-AM$^+$ 3T3 was measured with a confocal microscope (Eclipse Ti, $40\times$ objective; CFI Apo Lambda S LWD 40XWI, NA 1.25; Nikon), before a single FB was bleached with a 561 nm laser pulse (5 s, 2.5 mW). Subsequently, fluorescence images were taken every 15 s over 4 min and fluorescence intensity measured. Fluorescence intensities were converted into values between 1 (fluorescence intensity before bleaching) and 0 (fluorescence intensity immediately after bleaching).

## Optical mapping

Mice were killed by cervical dislocation 17 days after CI (i.e. 14 days after MMLV injection), their chest opened and the heart removed. Hearts were placed in warm (37°C) followed by cold (4°C) heparin-containing (10 units mL$^{-1}$) physiological saline solution, containing (in mM): 140 NaCl, 6 KCl, 10 HEPES, 10 glucose, 1.8 $CaCl_2$ and 1 $MgCl_2$, pH titrated with 1 M NaOH to 7.4, $300 \pm 5$ mosmol L$^{-1}$. The aorta was cannulated and the heart attached to a portable Langendorff system for dye loading and transfer to the measurement system. Voltage-sensitive dye (15 μL of 1.4 mM Di-4-ANBDQPQ; CytoVolt2; CytoCybernetics, Buffalo, NY, USA) was loaded in 0.4 μL increments over ~6 min for optical mapping. During dye loading flow was maintained at 1.5 mL min$^{-1}$; therefore, the dye was diluted in a final volume of 9 mL to a final concentration of 2.3 μM.

Once transferred to the measurement system (Hugo Sachs IH-SR System; Hugo Sachs Electronik, March-Hugstetten, Germany), hearts were perfused at a constant pressure of 60 mmHg. Spring-loaded Ag/AgCl electrodes were placed on right atrium and left ventricular apex to provide a pseudo-ECG. An additional concentric electrode was placed on the left ventricular at 10 o'clock relative to the scar for pacing. ECG, perfusion pressure, flow rate, temperature, camera exposure and electrical stimulation were recorded using ISOHEART software

(Hugo Sachs Electronik). Subsequent data analysis was performed using Matlab (Mathworks, Natick, MA, USA)

Hearts were illuminated with the red LED of a Light-crafter 4500 projector (Texas Instruments, Dallas, TX, USA) via an excitation filter (ET Dualband Emitter DAPI/Texas RED; Chroma Technology, Bellows Falls, VT, USA) and imaged with a CMOS (acA720-520um; Basler, Ahrensberg, Germany) via a 0.255 × telecentric lens (TDC-0255-70; Lensation GmbH, Karlsruhe, Germany) and an emission filter (ET585/50M+800/200M, Chroma Technology) at ~480 frames per second. The camera was controlled using Pylon Viewer (Basler).

Hearts were positioned so that the CI scar was clearly visible, at the same time as ensuring remote tissue was included in the field of view to monitor pacing. The ventricle was then stimulated at 7 Hz and optical mapping performed to assess activation patterns.

Optical mapping recordings were analysed using a custom-designed Matlab program (available from authors upon request). Optical mapping records were loaded and the background removed. The signal was then inverted and a pixel-by-pixel normalization performed. Pixel binning (4 × 4 pixels) and a spatial filter (3 × 3 pixel Gaussian, sigma = 0.5) were applied before averaging across multiple action potentials. Activation time maps were generated by selection of a seed pixel and performing cross-correlation of all pixels relative to the seed. A line was drawn on the activation time map from the location of the pacing electrode to the centre of the scar. The values along this line were used to assess the heterogeneity of conduction into the scar, using the technique to quantify image heterogeneity described by (Brooks & Grigsby, 2013).

### Statistical analysis

Statistical analysis was performed using Prism, version 10 (GraphPad Software Inc., San Diego, CA, USA). Data are depicted as the mean ± SD. Differences between two means were evaluated using an unpaired Student's *t* test. Differences between multiple means were tested by one-way analysis of variance (ANOVA) and *post hoc* Tukey's test. Electrophysiological data were analysed using Fisher's exact test. $P < 0.05$ was considered statistically significant difference.

### Results

#### *In vitro* transduction of 3T3 cells with MMLV/MNP complexes

Using different virus concentrations, we first evaluated the transduction efficiency of MMLV-mCherry (Fig. 1A) in 3T3 cells. After overnight incubation of cells with 500 VG per cell, 38.1% ($n = 4$) expressed red fluorescence

(mCherry$^+$) (Fig. 1*B* and *C*). At lower virus doses, significantly lower transduction rates were observed, with only 6.3% at 100 VG per cell ($P < 0.001$ *vs.* 500 VG per cell data) and 16.9% at 250 VG per cell ($P < 0.001$, $n = 4$ each) of mCherry$^+$ cells, respectively. A further increase of the virus dose to 1000 VG per cell) did not yield a significantly higher transduction rate (42.3%, $P = 0.332$, $n = 4$).

In the past, we increased the transduction efficiency of different EC lines by complexing lentivirus with MNP to allow magnetic steering (Trueck et al., 2012; Vosen et al., 2016). We tested this approach using the MMLV in combination with SoMag5-MNP, which consists of a magnetite core and a silica iron oxide shell (Mykhaylyk et al., 2012). Freshly generated MMLV/MNP complexes were added to cultured 3T3 cells at a dose of 500 VG per cell + 500 fg Fe/VG for 30 min ($n = 4$) (Fig. 1*D*). We found that the transduction efficiency without applying a magnetic field was 41.7%, which is not dissimilar from that achieved with overnight transduction with the MMLV alone (38.1%, $P = 0.799$, dose 500 VG per cell; see above). Upon magnetic field application for 30 min, the transduction rate was doubled to 80.5% ($P < 0.001$, $n = 4$). Next, we assessed whether the MMLV-mCherry/SoMag5 complexes were entirely absorbed by the cells after magnet-assisted transduction by transferring the supernatant to fresh 3T3 cells. After 30 min of incubation during magnetic field application, 17.6% ($n = 4$) of 3T3 cells were mCherry$^+$, indicating that residual complexes were reduced but still present in the supernatant (Fig. 1*D*). We therefore assumed that a lower MMLV concentration could provide similar FB transduction rates. MMLV concentration within the complexes was reduced to 250 VG per cell + 500 fg Fe/VG (Fig. 1*E* and F) and the percentage of mCherry$^+$ 3T3 cells remained very similar (81.2%, $n = 4$) after magnet-assisted transduction (30 min); still, 8.7% of fresh 3T3 were transduced by the supernatant. Without MNP/magnet support, reducing the MMLV dose from 500 VG per cell to 250 VG per cell decreased transduction efficiency by more than 50% after overnight incubation ($P < 0.001$, $n = 4$) (Fig. 1*D* and *F*).

To minimize potential adverse effects of an *in vivo* application, we also investigated a reduction of the MNP concentration. To do so, 250 VG per cell MMLV-mCherry + 250 fg Fe/VG SoMag5 complexes were added to 3T3 cells, and the magnetic field was applied for 30 min ($n = 4$). Furthermore, 72.9% of 3T3 cells expressed mCherry, which was not significantly different from the level attained with the higher Fe/VG SoMag5 concentration (Fig. 1*G*). We also examined whether the duration of magnetic field application influenced transduction rates and found that even 10 min were sufficient to achieve a 77.7% transduction rate (250 VG per cell + 250 fg Fe/VG, $n = 4$). Transduction efficiency was not significantly enhanced by prolonged exposure (30 min: 72.9%, 45 min: 75.0%, and 60 min: 71.8%, each $n = 4$) (Fig. 1*H*).

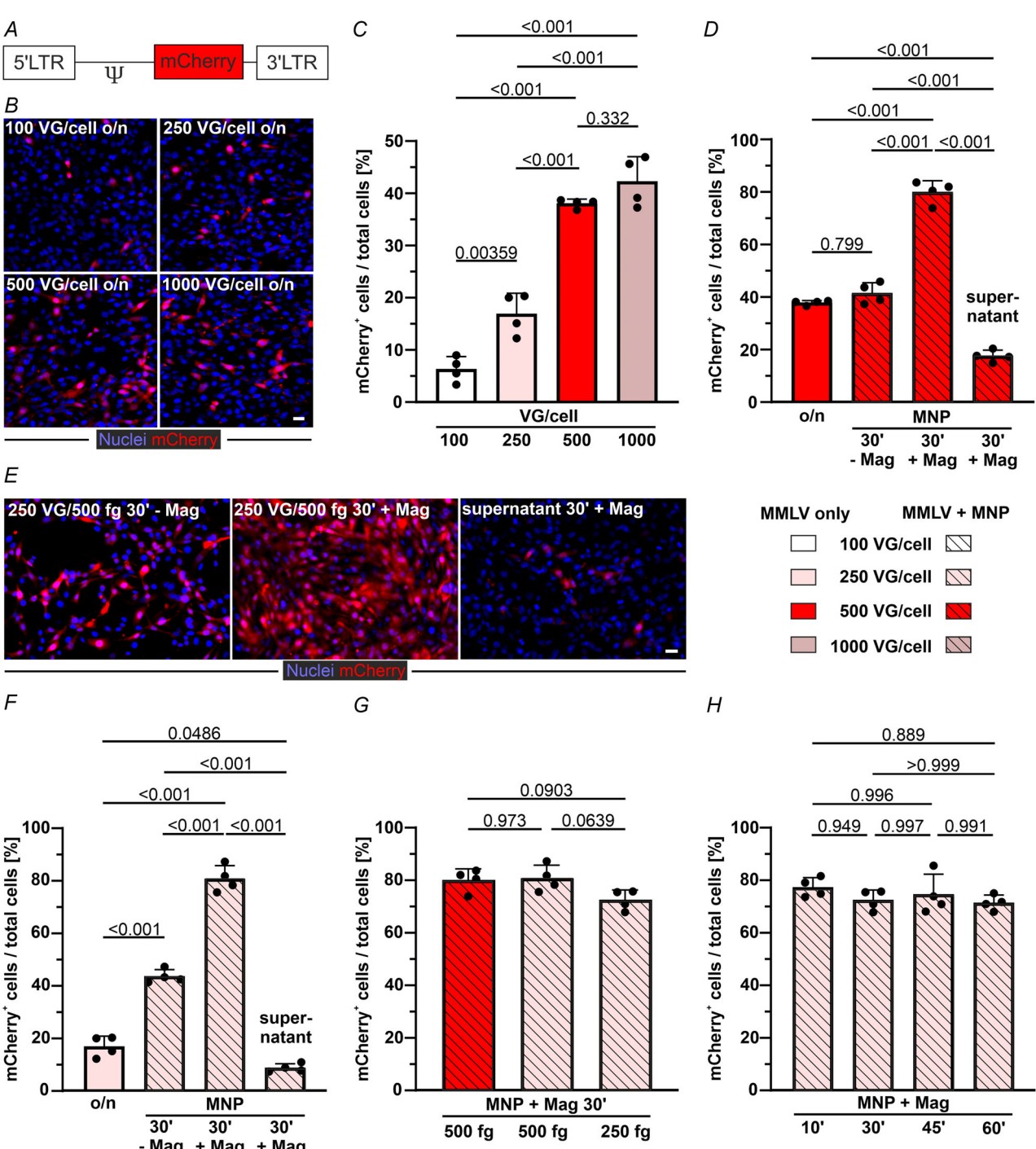

**Figure 1. MMLV-mCherry transduction of 3T3 cells *in vitro***
*A*, scheme of the MMLV-mCherry construct. *B*, microscopic images of mCherry+ 3T3 cells 3 days after overnight (o/n) incubation with different concentrations of MMLV (magnification: 20×, scale bar = 20 μm). *C*, percentage of transduced cells/ total 3T3 cells. *D*, percentage of mCherry+ 3T3 cells /total 3T3 cells 3 days after treatment with MMLV/MNP complexes [500 VG per cell + SoMag5-MNP (500 fg Fe/VG)] with (30 min, + Mag) or without magnetic field application (– Mag). After the first transduction, new cells were exposed to the supernatant with magnetic field application. *E*, representative microscopic images (magnification: 20×, scale bar = 20 μm) of mCherry+ 3T3 cells 3 days after transduction with 250 VG per cell + SoMag5-MNP 500 fg Fe/VG. *F*, quantification of the transduced cells. *G*, comparison of the transduction efficiency of MMLV-mCherry/SoMag5-MNP complexes with different virus and MNP concentrations after 30 min magnetic field application. *H*, impact of the duration of magnetic field application on transduction efficiency [250 VG per cell + SoMag5-MNP (250 fg Fe/VG)]. All data are expressed as the mean ± SD (*n* = 4 for each group); statistical evaluation by ANOVA and *post hoc* Tukey's test.

These experiments demonstrate that MMLV application can achieve very good transduction rates of FB. The combination of MNP and magnetic steering increased transduction efficiency by factor 4.6, enabling a reduction in the MMLV dose (250 VG per cell) and magnetic field exposure time (10 min).

### Formation of functional gap junctions following MMLV-mediated gene transfer in 3T3 cells *in vitro*

To assess whether MMLV-based transduction of 3T3 cells can have a functional impact, we tested the overexpression of the gap junction protein Cx43 using a virus coding for both mCherry and murine Cx43 (MMLV-Cx43) (Fig. 2*A*). In this construct, the two genes are connected by the GSG-P2A linker peptide (Niemann et al., 2022), which is cleaved during protein synthesis, leaving Cx43 with a short P2A tag of 21 amino acids allowing the direct identification of overexpressed Cx43. Upon overnight transduction of 3T3 cells with MMLV-Cx43, 23.7% of the cells showed mCherry expression. Complexation with SoMag5-MNP (250 VG per cell MMLV-Cx43 + 500 fg Fe/VG SoMag5) and 30 min magnetic field application significantly ($P < 0.001$, $n = 4$ each) increased the transduction rate (by a factor of 3) to 72.6% (Fig. 2*B*), similar to that with MMLV-mCherry.

Next, we investigated Cx43 protein expression by immunofluorescence and WB. After transduction with the MMLV-mCherry control virus, as expected, very few cells were Cx43$^+$ because 3T3 cells express endogenous Cx43 at low levels. Upon MMLV-Cx43 transduction, many 3T3 cells were positive for both Cx43 and P2A (Fig. 2*C*). WB analysis showed that Cx43 protein content did not differ significantly between untreated cells (WT) and MMLV-mCherry transduced cells, whereas Cx43 expression in MMLV-Cx43 transduced 3T3 cells was significantly increased two-fold ($P = 0.00157$ and $P = 0.0283$, respectively, each $n = 4$) (Fig. 2*D*).

Upon loading the cells with the gap junction-permeable dye Calcein-AM, we assessed Cx43 functionality by FRAP. When individual cells within a cell clusters were bleached with a laser light, the Calcein-AM influx from neighbouring cells (Fig. 2*E* and *F*) was measured over 4 min. Fluorescence recovery was significantly faster (five-fold higher after 2 min) (Fig. 2*E*) in MMLV-Cx43 transduced 3T3, compared to MMLV-mCherry transduced cells, confirming the formation of functional Cx43 gap junctions.

### Strongly increased transduction efficiency after MNP/magnet-based MMLV transduction of cardiac eFB *in vitro*

Next, the potency of MMLV to transduce cardiac FB was tested *in vitro* because the aim was to improve the transduction of scar FB *in vivo*. We took advantage of eFB because these can be easily cultured and expanded *in vitro*. In analogy to the experiments with 3T3 cells, the highest transduction efficiency (60.8%) of eFB was observed using MMLV-mCherry (250 VG/cell + 500 fg Fe/VG SoMag5-MNP) complexes in combination with a 30 min magnetic field application (Fig. 3*A* and *B*). The use of MMLV/MNP complexes without a magnet for 30 min (42.2%, $P < 0.001$) or overnight yielded significantly lower transduction rates (24.6%, $P < 0.001$, each $n = 5$). Residual virus activity was still observed in the supernatant following combined complex + magnetic field application (transducing 12.1% of previously non-exposed eFB, $n = 5$) (Fig. 3*A* and *B*). An increase of the MMLV amount (500 VG per cell), or a decrease of the MNP concentration (250 fg Fe/VG), did not significantly alter transduction rates (63.0% and 66.9%, respectively, each $n = 5$) (Fig. 3*C*). The transduction efficiency in eFB of MMLV-Cx43 alone or complexed with SoMag5-MNP (Fig. 3*D*) resembled the results obtained with MMLV-mCherry. Therefore, for all the following experiments, complexes consisting of 250 VG/cell + 250 fg Fe/VG were applied to minimize possible adverse side effects of the protocol in subsequent *in vivo* work.

Given that eFB express endogenous Cx43, we tested the degree of overexpression following MMLV-Cx43 transduction. Immunofluorescence indicated successful transduction by showing a strong P2A signal (Fig. 3*E*), which was confirmed in subsequent WB analyses. Although Cx43 levels were comparable in untreated and MMLV-mCherry/MNP (+ Magnet) eFB ($P = 0.670$), a two-fold increase was observed in MMLV-Cx43-transduced cells compared to both control groups ($P = 0.00300$ and 0.00692, respectively, $n = 3$ each) (Fig. 3*F*).

In conclusion, MMLV-Cx43 can transduce both 3T3 cells and eFB, analogous to MMLV-mCherry. After complexation with MNP, transduction efficiency can be increased substantially (three- to four-fold) by magnetic targeting to roughly double Cx43 expression in affected cells.

### Injection of MMLV/MNP complexes into the cardiac scar *in vivo* yields prominent transduction of resident cardiac FB

Because reproducible transmural myocardial lesions are critical for comparing analyses across the different study groups, we used CI as an injury model. Given the encouraging transduction results obtained *in vitro*, we injected $6.25 \times 10^7$ VG MMLV + 15.625 µg Fe SoMag5 (i.e. ∼250 Fe per VG) suspended in HBSS$^{++}$ (5 µL), as described in the Methods section. According to our *in vitro* data, this MMLV/MNP dose is sufficient to target at

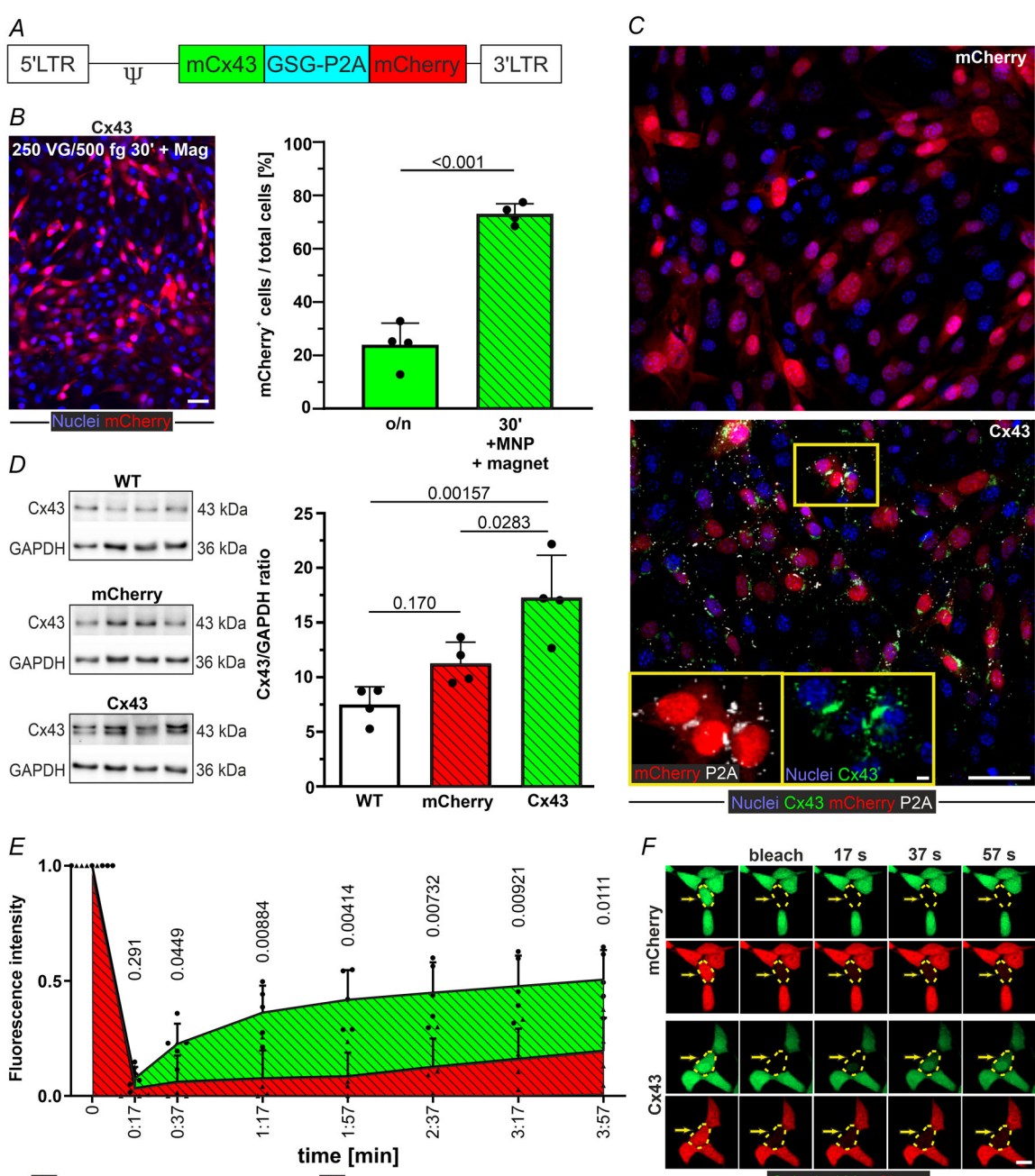

**Figure 2. Gap junctions in MMLV-Cx43 transduced 3T3 cells**
*A*, scheme of the MMLV-Cx43 construct. *B*, mCherry expression in 3T3 cells 3 days after magnet-assisted (30 min) transduction with MMLV-Cx43/MNP complexes (left, microscopic image: magnification: 20×, scale bar = 20 μm); transduction efficiency of MMLV-Cx43 in 3T3 cells: MMLV-Cx43 (250 VG per cell) was either applied overnight (o/n) or as MMLV/MNP complexes (250 VG per cell + 500 fg Fe/VG SoMag5-MNP) for 30 min with magnetic field application (right, *n* = 4 each). *C*, confocal images of transgene expression (mCherry ± Cx43) 3 days after transduction with MMLV/MNP + magnet-mediated transduction (green: Cx43, white: P2A, blue: nuclei, red: mCherry; magnification: 40x, scale bars = 20 μm; insets: 4 μm). *D*, WB-based assessment of Cx43 protein expression in wild-type (WT) and MMLV-mCherry or MMLV-Cx43 treated 3T3 cells (*n* = 4 each), original blot (left), quantitation of the blot (right). *E* and *F*, fluorescence recovery after photobleaching (FRAP) analysis after magnet-assisted transduction with the different viruses (confocal maximum projection, green: Calcein-AM, red: mCherry; magnification: 40x, scale bar = 20 μm; *n* = 4 (MMLV-mCherry, single data points are marked as filled dots) and *n* = 5 (MMLV-Cx43, single data points are marked as filled triangles). All data are given as the mean ± SD; statistical evaluation by an unpaired Student's *t* test (*B*), ANOVA and *post hoc* Tukey's test (*D*) and multiple *t* test (*E*).

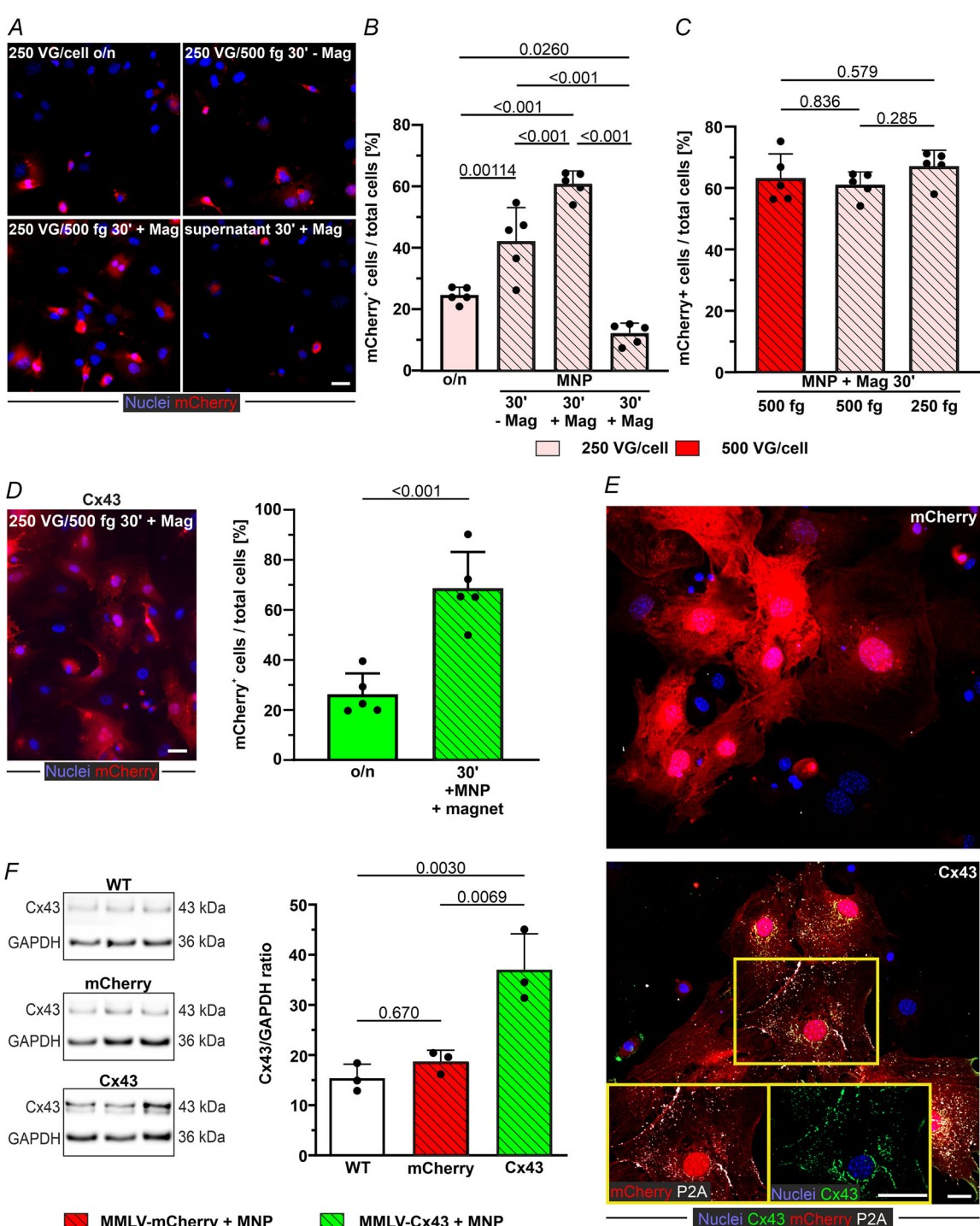

**Figure 3. MMLV-mediated Cx43 overexpression in eFB**

*A*, transduction efficiency of eFB 3 days after MMLV-mCherry incubation (250 VG per cell). MMLV exposure was carried out either with MMLV alone overnight or complexed with SoMag5-MNP (500 fg/VG), with or without magnetic field application for 30 min; after that, the supernatant was added to non-treated eFB (*n* = 5 each). Representative microscopic images of mCherry transduced eFB (magnification: 20x, scale bar = 20 μm) and their quantitation (*B*). *C*, comparison of transduction efficiency using different MMLV-mCherry/SoMag5-MNP concentrations in eFB (*n* = 5 for each group). *D*, transduction efficiency after application of MMLV-Cx43 alone (250 VG per cell) overnight (o/n) or complexed with SoMag5 (500 fg Fe/VG) for 30 min + magnetic field application (representative microscopic image shown) in eFB (each *n* = 5, scale bar = 20 μm). *E*, Cx43 expression in eFB 3 days after incubation with MMLV (confocal images: green, Cx43; white, P2A; blue, cell nuclei; red, mCherry;

magnification: 40×, scale bars = 20 μm). *F*, WB analysis of Cx43 protein expression in untreated eFB (WT) and eFB 3 days after transduction with MMLV-mCherry/MNP or MMLV-Cx43/MNP (each 250 VG per cell, 500 fg Fe/VG, *n* = 3 each). Data are given as the mean ± SD. Statistical evaluation by one-way ANOVA and *post hoc* Tukey's test (*B*, *C* and *F*) and an unpaired Student's *t* test (*D*).

least $2.5 \times 10^5$ cells in the lesioned tissue. As reported in our earlier work (Ottersbach et al., 2018), the rod magnet was positioned with a micromanipulator at a distance of 5 mm above the surface of the heart during the injection and for 10 min after that. MMLV complexes were injected 3 days after CI, a time at which resident FB are activated and known to peak in the proliferation rate, which is a prerequisite for successful MMLV transduction (Fu et al., 2018; Ramanujam et al., 2016). Mice were killed, and the hearts harvested for analysis 17 days after initial CI, which is 14 days after the consecutive MMLV injection. Prominent transduction of the scar area could be observed macroscopically based on the clearly visible mCherry expression (Fig. 4*A*).

Successful transduction was further illustrated by the overlap of red fluorescence (Fig. 4*B*, upper) MNP, which were visualized using Prussian blue staining (Fig. 4*B*, lower). In cryosections of the heart, substantial numbers of mCherry$^+$ cells (20,300 ± 9460 cells, *n* = 3) (Fig. 4*C*, right column) were seen in the scar area 14 days after MMLV injection. Apart from the absolute number of transduced cells, their distribution in the target area was also informative: In the central region of the intracardiac scar injections, in which the MMLV/MNP complexes were released and locally enriched by magnetic field application, 19.1% of cells were mCherry$^+$, whereas, at the margins, only 5.3% of cells were mCherry$^+$. This probably reflects the lower magnetic field strength and the lower concentration of MMLV/MNP complexes in the periphery of the CI scar. On average, 10.8% of cardiac scar cells in the injection area were mCherry$^+$ (Fig. 4*C*, left column).

We next characterized the identity of transduced cells, specifically the different types of FB in the cardiac scar, using immunofluorescence staining for various markers 17 days after CI (i.e. 14 days after MMLV injection). PDGFR$\alpha$, a general FB marker, was used in conjunction with $\alpha$SMA, which labels activated FB that should be present in the scar at this time. We found that 90% of mCherry$^+$ cells were FB (PDGFR$\alpha^+$), of which 41.5% were activated FB ($\alpha$SMA$^+$) (Fig. 4*D* and *F*).

Although MMLV only transduce proliferating cells, and CM do not proliferate in the adult heart, we thoroughly searched for mCherry$^+/\alpha$-actinin$^+$ cells in the scar, as well as in remote tissue. Although $\alpha$-actinin$^+$ CM were visible in the border zone of the lesion, only remnants without nuclei were present in the fibrous scar. Importantly, we did not detect any mCherry$^+/\alpha$-actinin$^+$ CM or non-CM, neither in the border zone of the lesion, nor in the remote

myocardium, excluding off-target effects of the virus on CM (data not shown).

MMLV has been reported to strongly transduce EC (Ramanujam et al., 2016) and we tested this aspect *in vitro*. We observed a transduction rate of 69.1% in C166 cells, using MMLV/MNP complexes in combination with a 30 min magnetic field application (data not shown). By contrast, only a negligible proportion of mCherry$^+$ EC (3%) was detected in heart sections after MMLV injection. There may be two explanations for this observation: proliferating EC were transduced at the time of injection but did not survive until the analysis at 2 and 8 weeks post-injection, and/or there were few proliferating EC, that had survived CI at the time of injection precluding transduction with MMLV (Fig. 4*E* and *F*).

It is known that activated resident or immigrating FB within the developing scar become senescent Tcf21$^+$ matrifibrocytes over time (Fu et al., 2018). Therefore, we also co-stained for Tcf21 and PDGFR$\alpha$. Quantitative analysis showed that again >90% of mCherry$^+$ cells were FB and that a large proportion of these cells (>70%) also expressed Tcf21, indicating that they already started to transdifferentiate to matrifibrocytes (Fig. 5*A* and *B*).

Because injection of MMLV might trigger host immune responses, corticoid treatment was prophylactically applied during the first 10 postoperative days (see also Methods section). Histological analysis of leukocyte infiltration of the hearts (17 days after CI) (i.e. 14 days post virus injection) was performed. There was no significant difference in the number of CD45$^+$ cells in the scar region between CI-MMLV injected and not injected CI control mice (Fig. 5*C*, left). Although few CD45$^+$ cells were located next to mCherry$^+$ cells, none were CD45$^+$/mCherry$^+$. Also long-term (59 days after CI) (Fig. 5*C*, right), there were still individual CD45$^+$ cells present in the scars of both non-transduced control and MMLV-mCherry injected mice. However, their number was very low (Fig. 5*C*) and no significant difference was detected between groups.

## MMLV/MNP-based Cx43 overexpression *in vivo* significantly reduces the incidence of post-infarct VT in the short- and long-term

To investigate the functional impact of an MMLV/MNP-based gene therapy in the infarcted heart *in vivo*, MMLV-Cx43/MNP complexes were injected and local enriched using magnetic steering (CI-Cx43). Very similar to the MMLV-mCherry/MNP injected

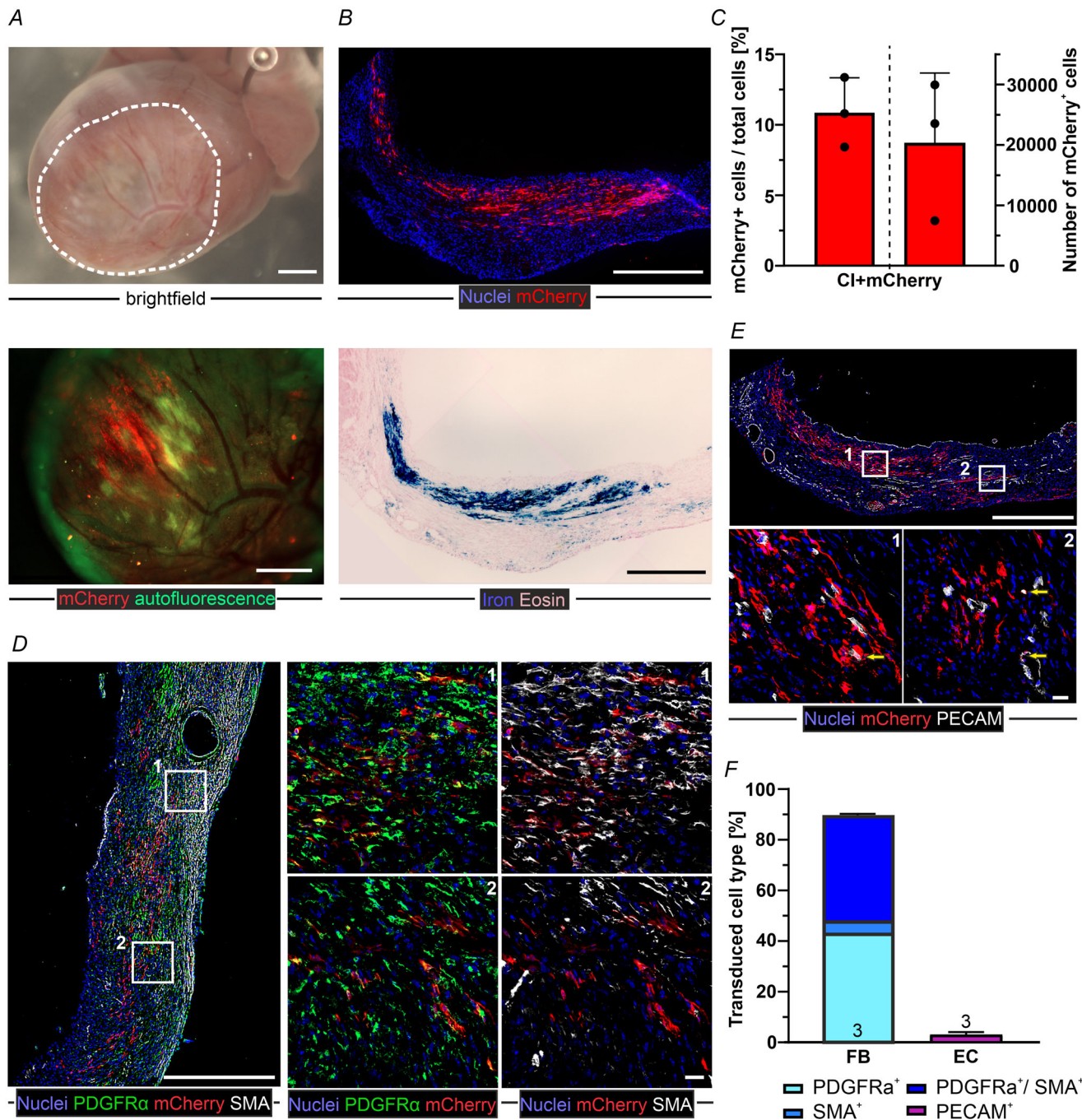

**Figure 4. MMLV/MNP based targeting of the scar tissue in mouse hearts *in vivo* (17 days after CI) (i.e. 14 days after MMLV injection)**

*A*, macroscopic (upper) and fluorescence (lower) images of a harvested heart following CI and MNP/magnet assisted intramyocardial MMLV-mCherry injection (top: brightfield; magnification: 0.8×, white circle shows infarction area, bottom: fluorescence image; magnification: 1.6×, scale bars = 1000 μm). *B*, immunostaining for mCherry (upper picture) and Prussian blue iron staining for MNP (lower picture) mark the transduced tissue region in the cardiac scar (microscopic images, top: magnification: 20×, bottom: magnification: 6.3×, scale bars = 500 μm); in (*A*) and (*B*), different hearts are shown. *C*, quantitation of the percentage (left) and the total number (right, *n* = 3) of mCherry$^+$ cells in the targeted scar region. *D*, immunofluorescence stainings for mCherry, the (m)FB markers PDGFRα and αSMA, and *E*, the EC marker PECAM in the scar region of MMLV-mCherry injected hearts (*D*, left; *E*, upper: microscopic mosaic images; magnification: 20×, scale bar = 500 μm. *D* and *E*, detailed images: confocal images of sections of the scar; magnification: 40×, scale bar = 20 μm. Blue, cell nuclei; red, mCherry; green, PDGFRα; white, αSMA (*D*) and PECAM (*E*). *F*, quantitation of PDGFRα− and/or αSMA and PECAM expression of mCherry$^+$ cells (*n* = 3 each).

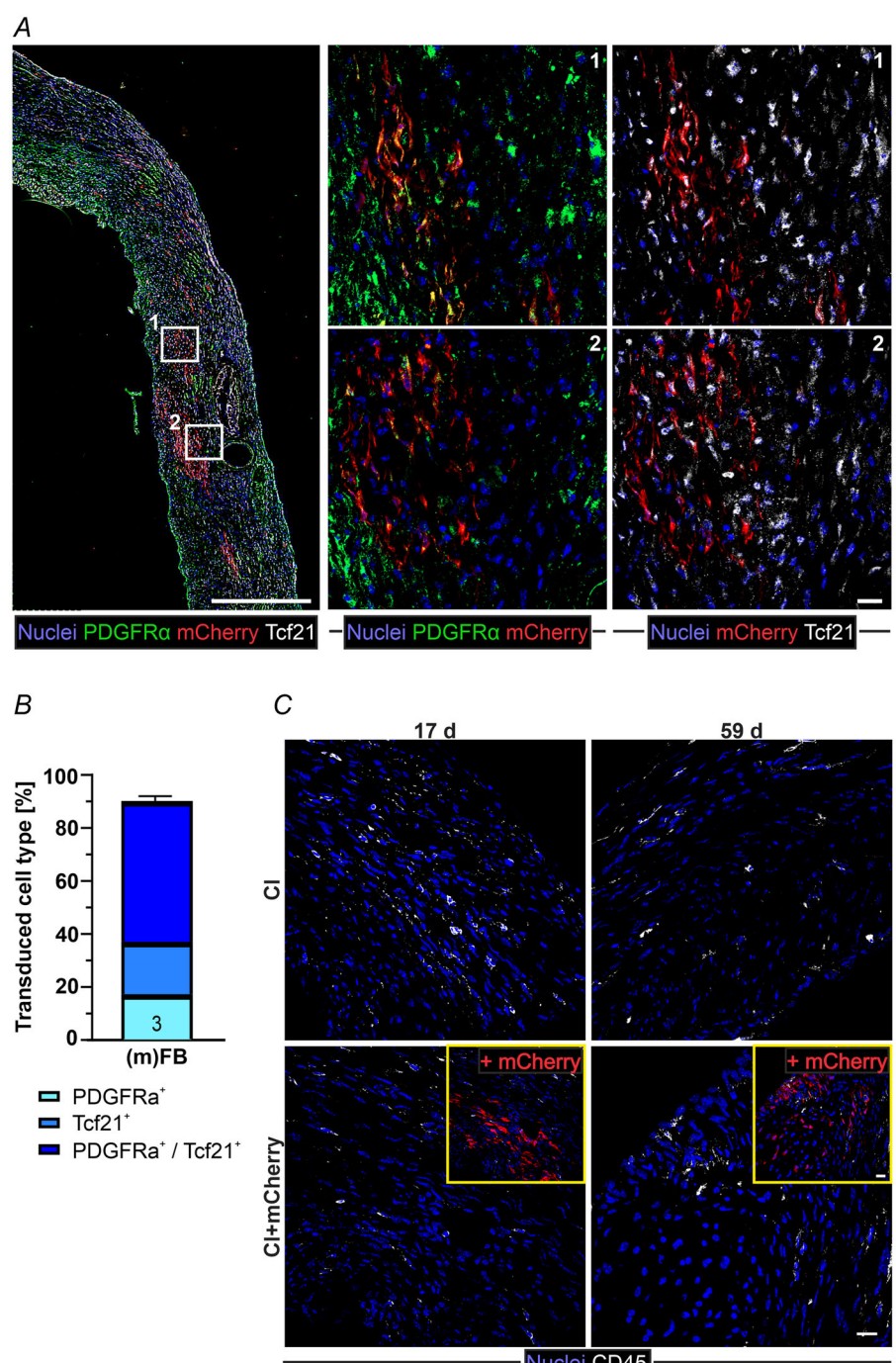

**Figure 5. Characterization of FB phenotype following *in vivo* (17 days after CI) (i.e. 14 days after MMLV injection), leukocyte infiltration at different time points (17 days and 59 days after CI)**
*A*, immunostaining against mCherry, PDGFR$\alpha$ and Tcf21 in the scar region of CI-mCherry hearts (blue, nuclei; red, mCherry; green, PDGFR$\alpha$; white, Tcf21. Left: microscopic mosaic image; magnification: 20×, scale bars = 500 µm. Right: confocal maximal projections; magnification: 40×, scale bars = 20 µm). *B*, quantification of PDGFRa[+], Tcf21[+] and PDGFRa/Tcf21[+] cells (*n* = 3 hearts each). *C*, immunostaining against mCherry and the immune cell marker CD45 in scar areas after CI (top) and CI-mCherry (bottom) hearts over time (left: 17 days; right: 59 days after CI w/o MMLV-mCherry injection 3 days after CI (blue, nuclei; white, CD45; insets: red, mCherry, confocal maximal projections; magnification: 40×, scale bars = 20 µm). Data are expressed as the mean ± SD.

hearts (CI-mCherry), mCherry expression and MNP particle distribution at 17 days post CI and consecutive MMLV-injection (3 days later) were substantial (Fig. 6*A*), with 8.2% of cells being mCherry$^+$ (range 2.4–15.6%, $n = 3$) (Fig. 6*B*, left) and a total number of transduced cells of 19 657 $\pm$ 4201 cells ($n = 3$) (Fig. 6*B*, right). Immuno-staining of heart sections showed only sporadic punctate Cx43 signal in cardiac FB in the scar area of CI-mCherry hearts (Fig. 6*D*, left). At 14 days post-MMLV-Cx43 treatment, strong Cx43 expression was observed in the transduced scar tissue; this was corroborated by the P2A signal (Fig. 6*D*, right, insets). Next, we assessed Cx43 gene expression in excised scar tissue by dPCR and found a significant increase in the CI-Cx43 group, compared to CI and CI-mCherry hearts ($P = 0.00965$ both, $n = 5$ each) (Fig. 6*C*), confirming the results of immunohistology.

Quantitative morphometry of the lesioned hearts showed that MMLV/MNP treatment did not alter scar volume (Fig. 7*A*) or epicardial surface area (data not shown) 17 days after CI. Echocardiography illustrated a significant reduction in anterior wall thickening and fractional shortening in all lesioned groups, compared to sham-operated animals ($n = 37$, $P < 0.001$), consistent with gross morphological findings post-mortem. There were no significant differences in cardiac pump function after isolated CI, ($n = 36$), CI-mCherry ($n = 43$), and CI-Cx43 ($n = 40$) (Fig. 7*B*).

Our earlier work indicated that Cx43 overexpression in cardiac scar tissue reduces post-MI VT incidence. Using electrophysiological testing *in vivo* (Fig. 7*C*), we assessed potential protective effects of Cx43 overexpression in the cardiac scar, using extra- and burst stimulus protocols. In CI-mCherry mice upon burst stimulation, ventricular capture, followed by a self-terminating VT period with A-V dissociation were observed in the vast majority of animals (representative example in Fig. 7*C*, upper two traces), whereas in the CI-Cx43 injected mice burst pacing with ventricular capture was followed by normal sinus rhythm after a short compensatory pause in most cases (see example in Fig. 7*C*, lower two traces). Burst pacing-induced VT incidence was significantly lower in CI-Cx43 hearts (36.8%, $n = 19$) compared to both CI (84.2%, $n = 19$, $P = 0.00694$) and CI-mCherry treated hearts (73.7%, $n = 19$, $P = 0.0489$) (Fig. 7*C*, right).

We next examined the potential mechanism(s) by which Cx43 overexpression in the cardiac scar region decreases the incidence of VT, using optical mapping in Langendorff-perfused hearts (Fig. 7*D*). We detected no significant difference in the average total activation time across the visible ventricular surface between CI-mCherry and CI-Cx43 mice (19.0 ms $\pm$ 5.9 ms vs. 15.8 ms $\pm$ 13.3 ms, $P = 0.68$, $n = 4$ or 5 mice per group, mean $\pm$ SD). However, electrical conduction is more homogenous in CI-Cx43 mice, as seen in the activation maps with less isochrones crowding at the border of the

scar and quantified by the heterogeneity of conduction into the scar region (0.036 ms $\pm$ 0.0074 ms vs. 0.019 ms $\pm$ 0.0094 ms, $P = 0.0222$, $n = 4$ or 5 mice per group, mean $\pm$ SD) (Fig. 7*D* and *E*).

Because retroviruses, such as MMLV, stably integrate into the host-cell genome, mice were also analysed 8 weeks after intramyocardial magnet-assisted virus transduction. Prominent mCherry expression was observed in MMLV-mCherry and MMLV-Cx43-injected scars (Fig. 8*A* and *C*). Quantification of mCherry$^+$ cells 59 days after CI (i.e. 56 days post-MMLV-Cx43 injection) revealed 8,515 $\pm$ 1,397 transduced cells in the scar region (Fig. 8*B*). Likewise, sustained Cx43 expression was still detected in MMLV-Cx43 treated hearts, as shown by immuno-fluorescence (Fig. 8*C*).

Next, we assessed heart function and electrical vulnerability by performing echocardiography (day 58) (Fig. 9*A* and *B*) and electrophysiological testing (day 59) (Fig. 9*C*) after CI and following MMLV injection (day 3 post CI). Similar to the earlier timepoint (Fig. 7*B*), fractional shortening (Fig. 9*B*) was significantly reduced in the CI groups compared to non-infarcted sham controls ($P < 0.001$). Importantly, even after a further 6 weeks, CI-Cx43 mice showed a significantly reduced VT incidence (down to 50%, $n = 20$) compared to both CI (82.6%, $n = 23$, $P = 0.0483$) and CI-mCherry mice (85%, $n = 20$, $P = 0.0407$), illustrating the preservation of the protective effect observed earlier (Fig. 9*C*).

We also analysed VT events and duration in both time points (17 and 59 days post CI, i.e. 14 and 56 days post MMLV injection). Although VT events per animal increased significantly over time in CI and CI-mCherry mice, it remained stable in the CI-Cx43 cohorts over time (Fig. 9*D*). VT duration (Fig. 9*E*) as well as VT frequency and basal heart rate (data not shown) were not significantly different between the groups at both time points.

## Discussion

Myocardial infarction and its primary sequelae (i.e. heart failure and ventricular arrhythmias) occur frequently, are associated with high morbidity and mortality, and impose a great personal and socioeconomic burden (Cao et al., 2024; Jentzer et al., 2023). Besides preventive measures, therapeutic strategies that target the cardiac scar to modulate the inflammatory response and mitigate cardiac remodelling could be promising treatment options. However, targeting the fibrous cardiac scar has proven very challenging so far, most probably because of its lower degree of vascularization (Pei et al., 2022) and the limited plasticity of resident FB (Venugopal et al., 2022). Classic AAV are very efficient in targeting and transducing CM. However, the various AAV serotypes

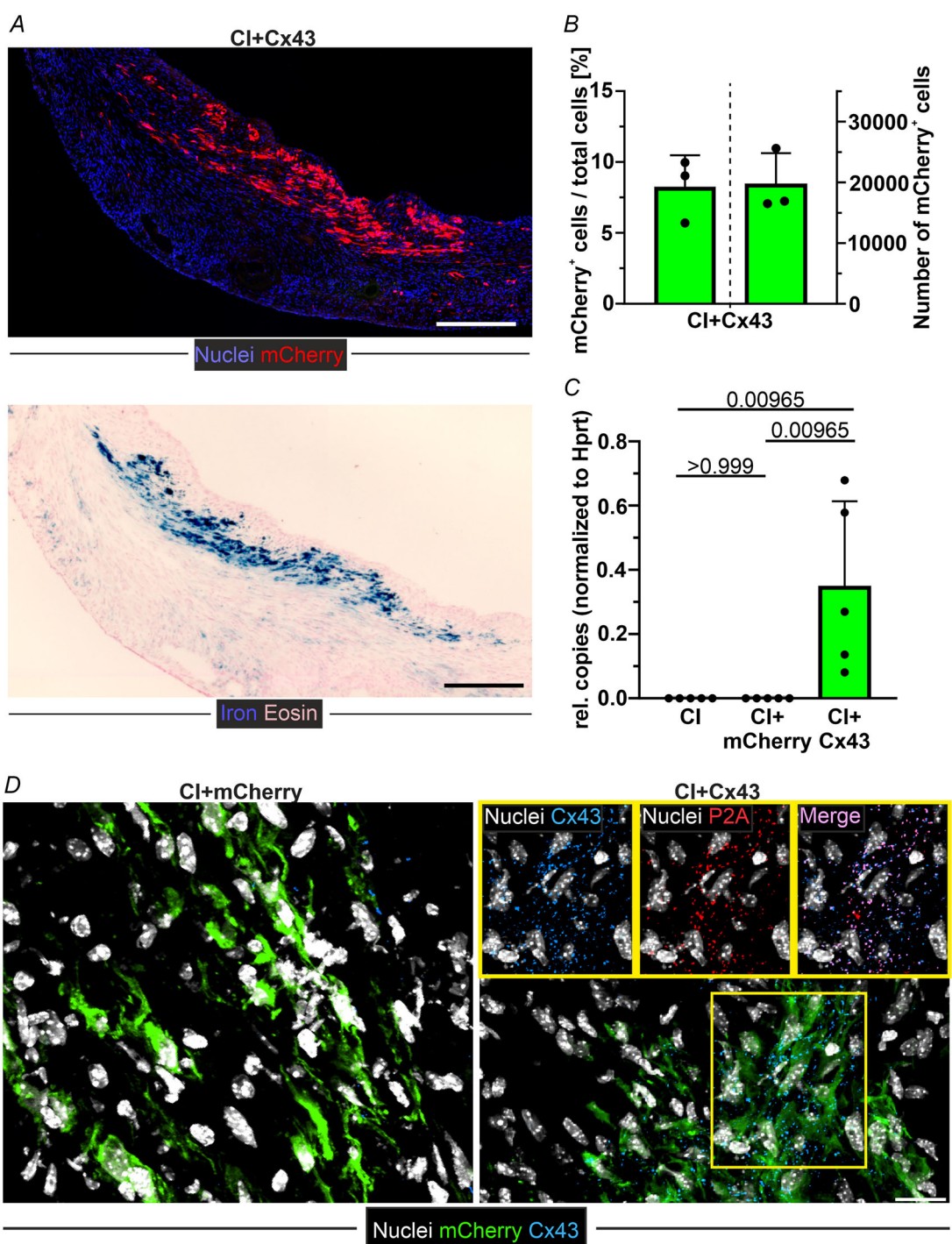

**Figure 6. MMLV/MNP-based Cx43 overexpression in the myocardial scar tissue *in vivo* (17 days after CI) (i.e. 14 days after MMLV injection)**

*A*, mCherry expression (upper picture) and MNP distribution (lower picture) in the scar area after MMLV/MNP injection (mosaic images, top: mCherry immunostaining; magnification: 20×, bottom: Prussian blue staining; magnification: 6.3×, scale bars = 250 μm). *B*, quantification of mCherry$^+$ cell percentage (left column) and number (right column) in the targeted scar area (*n* = 3). *C*, quantitation of exogenous Cx43 RNA in excised myocardial scar tissue via dPCR in MMLV-mCherry and MMLV-Cx43 hearts (*n* = 5 each). *D*, confocal images of immunostaining for mCherry, Cx43, and P2A in scar tissue of MMLV-mCherry and MMLV-Cx43-injected hearts (white, cell nuclei; green, mCherry; blue, Cx43; red, P2A; magnification: 40x, scale bars = 20 μm). All data are given as the mean ± SD. Statistical evaluation by one-way ANOVA and *post hoc* Tukey's test.

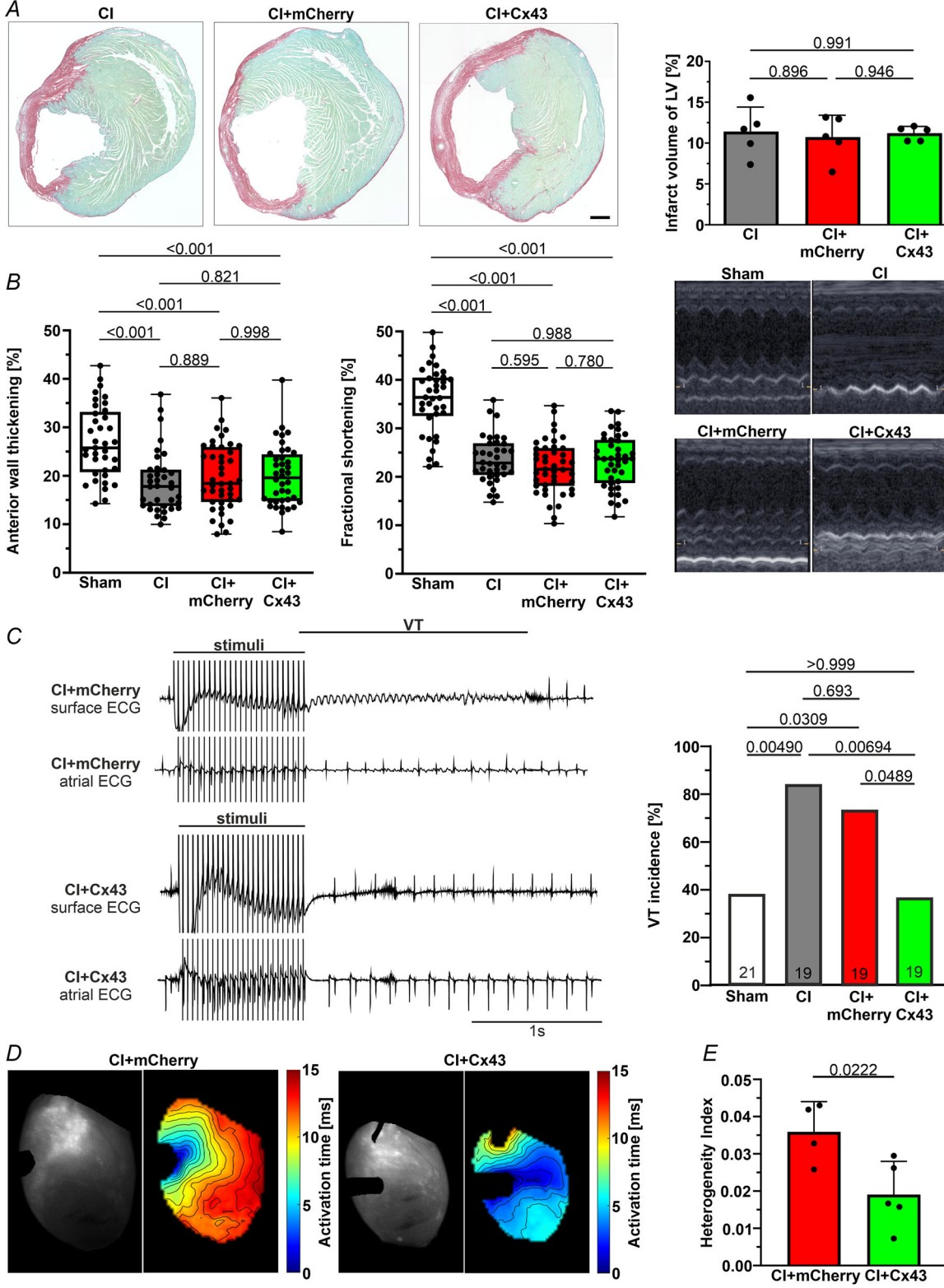

**Figure 7. Echocardiography, *in vivo* electrophysiological testing and optical mapping of murine hearts following CI and treatment with MMLV-mCherry or MMLV-Cx43 (17 days after CI) (i.e. 14 days after MMLV injection)**

*A*, scar size of hearts (*n* = 5 each); left: representative mosaic images of heart sections stained for Sirius red-Fast green (magnification: 3.2×, scale bar = 500 μm); right: infarct volumes determined by quantitative

morphometry (sections every 400 µm). *B*, echocardiographic analysis of anterior wall thickening (left) and fractional shortening (centre), with original M-mode images (right). *C*, left: representative surface and atrial ECG tracings from CI-mCherry (upper two tracings) and CI-Cx43 (lower two tracings) mice. Right: VT incidence. *D*, images of isolated Langendorff-perfused CI-mCherry or CI-Cx43 hearts during optical mapping (left) and activation time maps when pacing at 7 Hz (right). *E*, heterogeneity of conduction from the pacing site into the scar [$n = 4$ (MMLV-mCherry) and $n = 5$ (MMLV-Cx43)]. Data in (*A*) and (*E*) are presented as the mean ± SD. Data in (*B*) are presented as box plots with the whiskers showing all data points from minimum to maximum (Sham: $n = 37$, CI: $n = 36$, CI+mCherry: $n = 43$, CI+Cx43: $n = 40$). Statistical evaluation by by one-way ANOVA and *post hoc* Tukey's test (*A* and *B*), Fisher's exact test (*C*) and Student's *t* test (*E*).

have not shown strong tropism for FB, even if some reports have indicated transduction of cardiac FB by AAV (Nakano et al., 2024; Nieto et al., 2024; Piras et al., 2016).

Unfortunately, AAV transduction properties and cell type specificity *in vivo* remain unclear. Moreover, genome non-integrating virus types, such as AAV, have the drawback of being diluted in strongly proliferating cells, such as cardiac FB, in particular shortly after MI. This can be avoided by the use of retroviruses. We and others (Abouleisa et al., 2024) have tested lentivirus in the past to transduce the murine myocardial scar *in vivo*. However, the transduction rates of FB *in vitro* (25%) and, more so, *in vivo* were very poor because only 1.1–2.8% of FB in the infarcted mouse heart were targeted 2 weeks after injection could be transduced (Roell et al., 2018). Therefore, we tested MMLV, a retrovirus known to stably

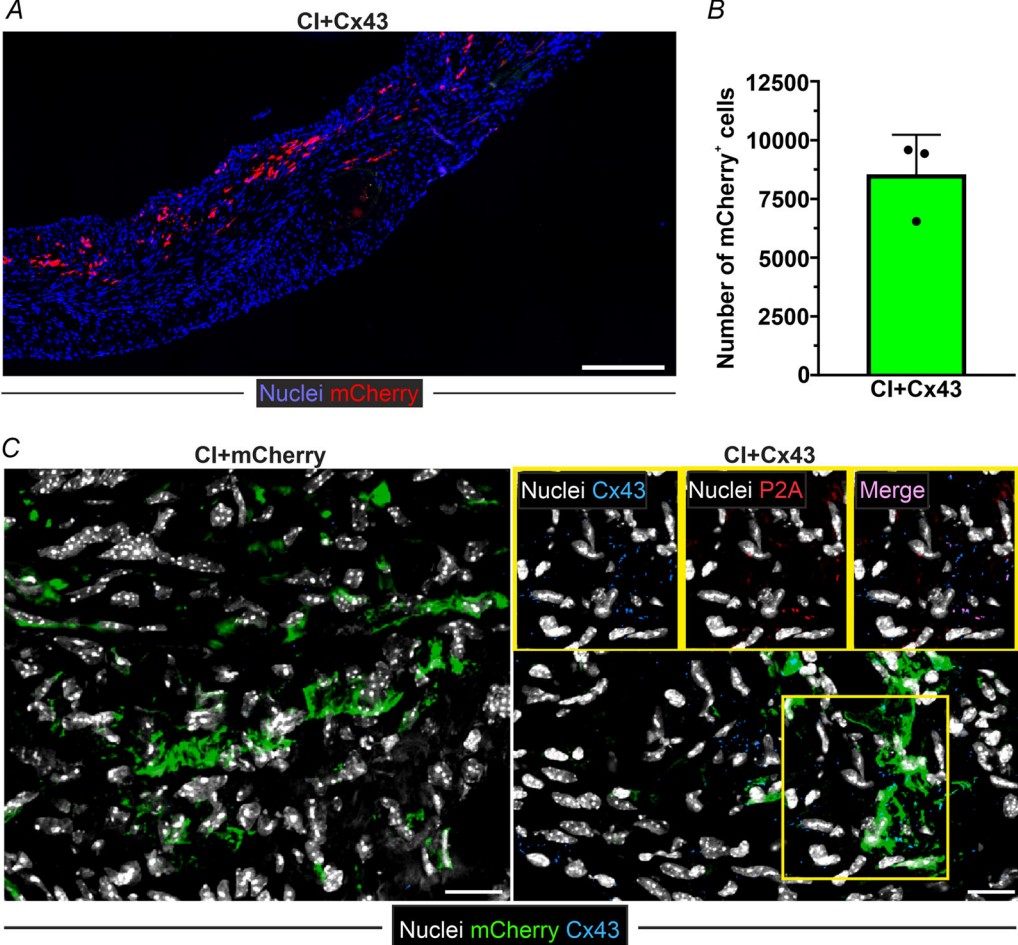

**Figure 8. MMLV/MNP-based gene expression in the myocardial scar *in vivo* (59 days after CI, i.e. 56 days after MMLV injection)**
*A*, fluorescence image showing mCherry expression in the scar area after MMLV/MNP injection (mosaic image. mCherry immunostaining; magnification: 20×, scale bar = 250 µm, data are given as the the mean ± SD). *B*, quantitation of number of mCherry[+] cells in the scar area ($n = 3$). *C*, confocal images of immunostaining for mCherry, Cx43 and P2A in scar areas of MMLV-mCherry and MMLV-Cx43-injected hearts (white, cell nuclei; green, mCherry; blue, Cx43; red, P2A; magnification: 40×, scale bars = 20 µm).

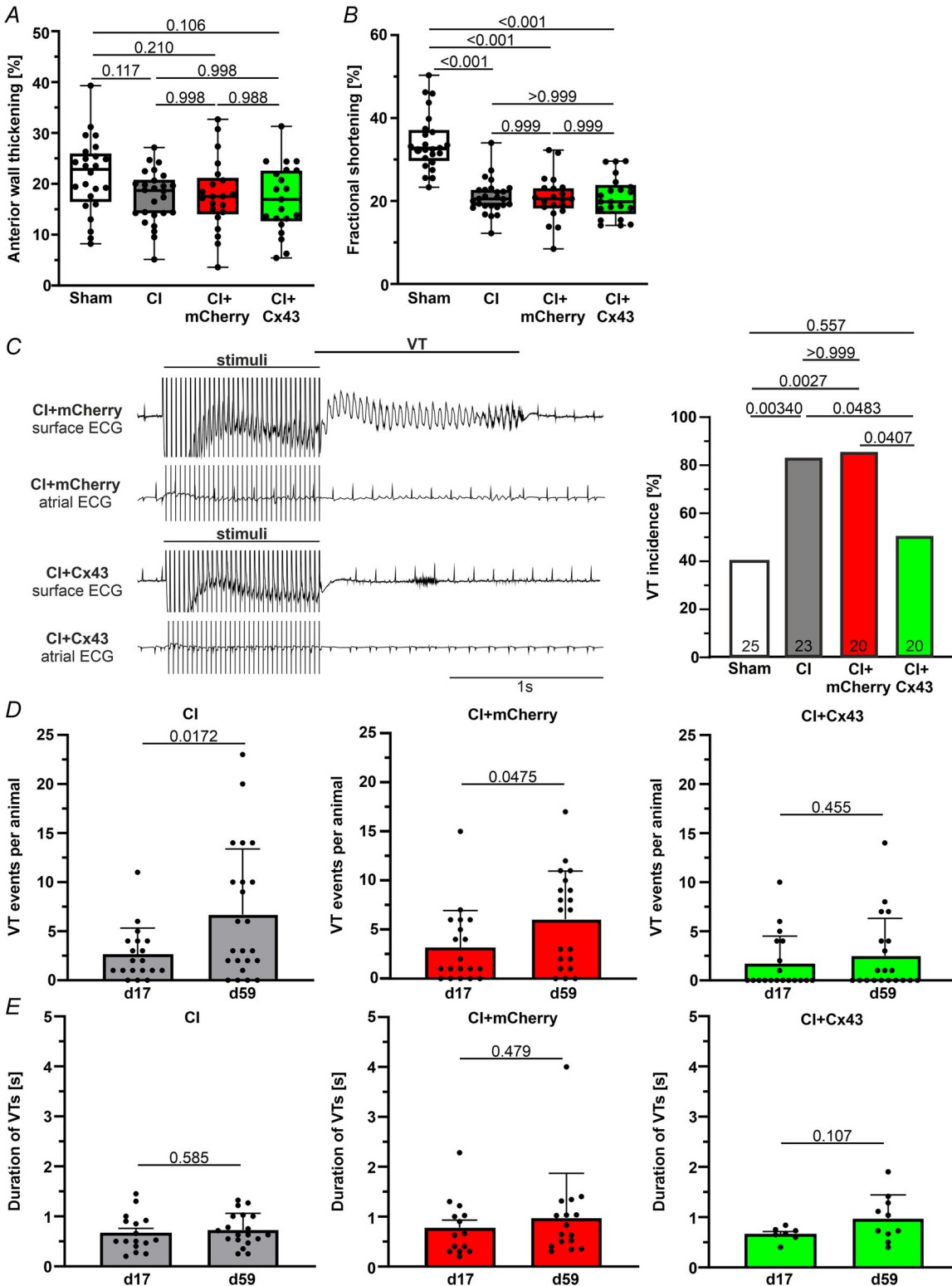

**Figure 9. Echocardiography and *in vivo* electrophysiological testing of mice 59 days after CI, that is 56d after MMLV injection**

*A*, echocardiographic analysis of anterior wall thickening. *B*, fractional shortening. *C*, left: representative surface and atrial ECG tracings from CI-mCherry (upper two tracings) and CI-Cx43 (lower two tracings) mice. Right: VT incidence. *D*, number of induced VT runs per animal during electrophysiological testing, comparing short- (17 days after CI) and long-term (59 days after CI) results in CI (left), CI-mCherry (middle) and CI-Cx43 (right) mice (MMLV

injection 3 days after CI). *E*, average VT duration. Data in (*A*) and (*B*) are presented as box plots with the whiskers showing all data points from minimum to maximum (Sham: *n* = 24, CI: *n* = 26, CI+mCherry: *n* = 21, CI+Cx43: *n* = 21). Statistical evaluation by one-way ANOVA and *post hoc* Tukey;s test (*A* and *B*) and Fisher's exact test (*C*). Data in (*D*) and (*E*) are expressed as the mean ± SD. Statistical evaluation using an unpaired Student's *t* test.

integrate into the genome of dividing host cells (Bukrinsky et al., 1993; Roe et al., 1993) as proof of concept. This was motivated by earlier work, showing good transduction of cardiac EC (70%) and FB (30%) (Ramanujam et al., 2016) after injection into the pericardial sac of newborn mice.

For our *in vitro* experiments, we used 3T3 cells and eFB, which are highly proliferative, can be expanded in cell culture, and resemble activated cardiac FB after MI (see also below). However, when testing MMLV-based overnight transduction of FB *in vitro*, the transduction rate was ∼20%, which is relatively modest. MMLV, complexed with MNP and exposed to a magnetic field that was co-applied for 30 min (Trueck et al., 2012), strongly increased the transduction rate of FB up to 80%, which is 4-fold higher than that achieved with an overnight incubation with MMLV only. We assessed reduced virus concentrations in light of possible immunogenic reactions upon *in vivo* injection. We found that halving the MMLV concentration (from 500 to 250 VG per cell) did not significantly affect the transduction rate. Because MMLV primarily transduce dividing cells, we reasoned that MMLV-based transduction *in vivo* should be optimal at the time point when the proliferation rate of resident FB is highest after cardiac injury. This has been reported to occur 2–4 days after left anterior descending artery ligation-induced MI during the post-ischemic remodelling process (Fu et al., 2018). The strategies employed to enhance the transduction of FB in the *in vitro* experiments were also utilized for the *in vivo* injections, specifically complexation of MMLV with MNP, combined with magnetic steering by positioning a rod magnet near the surface of the heart. This method helps to reduce the reflux of injected material through the injection channel and washout from the targeted tissue. SoMag5-MNP was used; these particles display a good magnetic moment and low cellular toxicity (Mykhaylyk et al., 2012) because they are externalized over time from the cells and subsequently excreted, primarily via the liver and kidneys (Ottersbach et al., 2018). Using this targeting strategy, we achieved solid and reproducible transduction of 10.8% of cells in the scar 17 days after CI, exceeding previous levels by an order of magnitude (Roell et al., 2018). Because the transduction rate of resident scar cells was even higher in the central area of the injection (>19%), modulation of the shape and strength of the magnet, the use of other types of MNP and/or altered virus concentrations may provide even better results. Scar FB have been found to undergo phenotypic changes because, for the first 10 days, they mostly present as activated FB ($\alpha$SMA$^+$/PDGFR$\alpha^+$), whereas, later, they become matrifibrocytes ($\alpha$SMA$^-$/Tcf21$^+$) (Fu et al.,

2018). Immunostaining for PDGFR$\alpha$, 17 days post CI (i.e. 14 days after MMLV injection), revealed that almost all mCherry$^+$ cells were FB, whereas staining for $\alpha$SMA and Tcf21 suggested that >70% of FB were in transition to activated FB and/or matrifibrocytes (Tcf21$^+$). That this process was still ongoing in ∼50% of the cells at the time of analysis was demonstrated by the fact that ∼50% of the FB still expressed $\alpha$SMA. Compared to earlier work (Fu et al., 2018), we detected a slower transition of activated FB to the matrifibrocyte stage, most probably as a result of differences in lesion type and the MMLV transduction. By contrast to previous reports and our *in vitro* data, EC in the scar were poorly transduced (Ramanujam et al., 2016). After MI in mice, revascularization starts between days 2 and 4 and progresses from the border zone to the centre of the lesion. Because new capillaries mostly develop from pre-existing EC in the lesioned area, and non-EC sources do not contribute significantly to the re-capillarization of the scar, only a few surviving EC will have been present in the target area 3 days after CI (i.e. at the time of MMLV injection) (He et al., 2017; Wu et al., 2021). We also did not detect a prominent immune response in the scar area 14 days after virus injection because only a few CD45$^+$ cells were detected and at levels that did not significantly differ from non-transduced CI control hearts. This could be a result of the use of a lower virus concentration than in previous studies (Fehse & Roeder, 2008; Song et al., 2019) or the administration of cortisone for ten days post-virus injection (Daseke et al., 2020; Madsen et al., 2023; Prabhu & Frangogiannis, 2016). However, even at 8 weeks post-treatment and without immunosuppressive therapy beyond the 10 day period, no obvious signs of increased inflammation in the cardiac scar area were observed. Regionally targeted transduction of cells within the scar by magnetically steered MMLV was confirmed by the lack of mCherry$^+$ cells in remote myocardium in any of the screened heart sections. Thus, the type of virus, the administration technique and the timing of gene therapy appear to be suitable for optimal transduction of cardiac scars.

Our experimental results further demonstrate that gene targeting of cardiac scars appears to be highly promising in terms of therapeutic potential. This was illustrated by the functional impact of overexpressing Cx43 in resident FB of the scar. The loss of CM and the reduction in Cx43 expression in and around cardiac scars give rise to impaired electrical conduction in this region, which contributes to the occurrence of VT after MI (Donahue et al., 2024). Our earlier work has shown that, in mice, post-MI VT primarily arises because of re-entry and that

even a low degree of Cx43 overexpression in the scar tissue decreases wave breaks and increases conduction velocity in both the border zone and the central scar region (Roell et al., 2007; Roell et al., 2018). In the present study, we demonstrate strong expression of Cx43 in FB transduced by MMLV-Cx43 and show the formation of functional gap junctions *in vitro* using FRAP. MMLV-based over-expression of exogenous Cx43 in the scar was confirmed by P2A expression. At 2 and 8 weeks following virus injection, a substantial reduction in VT inducibility by ∼50% was observed, similar to that reported previously upon grafting embryonic CM (Ottersbach et al., 2018; Roell et al., 2007). Although VT events per animal increased significantly over time in CI- as well as in CI-mCherry control animals, the number of VT events in CI-Cx43 animals remained stably low over time. Importantly, the postoperative survival rate was also very similar between CI-mCherry and CI-Cx43 mice, ruling out major Cx43-dependent adverse effects. Neither scar size, nor pump function was altered by MMLV-Cx43 transduction. Interestingly, in late-stage hearts, a lower number of transduced cells was observed because the cardiac scar is known to thin over time due to cellular loss and tissue expansion (Frangogiannis, 2021; Schiffer et al., 2025). This may affect the longevity of gene therapy, although the protective anti-VT effect was retained in our model for at least 8 weeks. Interestingly, such a protective effect was also observed when grafting Cx43 overexpressing eFB into CI (Schiffer et al., 2025). Optical mapping of isolated Langendorff-perfused hearts showed that the increased presence of Cx43 was associated with a reduction in the heterogeneity of conduction between healthy and scarred myocardium, which may explain the reduction in VT inducibility (Han et al., 2021). Under-lying cellular mechanism(s) may include heterocellular coupling, which has been reported for cardiac FB and CM in murine ventricular tissue after CI (Quinn et al., 2016) and coronary occlusion (Quinn et al., 2016; Rubart et al., 2018; Schultz et al., 2019). So far, we have not been able to validate the cellular nature of protective mechanism(s) unequivocally, given the structural and functional heterogeneity of the scar in mouse hearts and the technical limitations of using voltage mapping. In addition, assessment of clinical relevance calls for a large animal model of this interesting aspect.

In summary, we present a strategy to efficiently target and transduce cardiac scar tissue *in vivo*, using a combination of MMLV complexed with MNP and magnetic steering. This approach may pave the way towards novel treatment strategies for common post-MI sequelae such as heart failure and VT. It is doubtful that MMLV will be the ultimate shuttle of choice for large animals and humans, even with a change in virus packaging, because this type of retrovirus carries the risk of proto-oncogenic integration (Cavazza et al., 2013;

Huang et al., 2018). Interestingly, preliminary experiments in our laboratory indicate that lentivirus may be more efficient in transducing porcine cardiac FB in combination with MNP and magnetic field targeting *in vitro*. Therefore, testing different types of retroviruses in porcine and human cardiac tissue, along with MNP and probing these in combination with newly designed magnets for large animal testing, could be a promising strategy to enable effective gene therapy for cardiac scars or other organ systems (Azadpour et al., 2023; Mätlik et al., 2014; Shalaby et al., 2016) and to assess the utility of this approach for human patients.

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

## Additional information

### Data availability statement

The raw/processed data will be made available on request.

### Competing interests

The authors declare that they have no competing interests.

## Author contributions

T.M. performed complexation experiments (3T3, EC), surgeries, echocardiography, western blots, immunostainings and data analysis. M.S. isolated/transduced eFB and performed surgeries, electrophysiological testing, FRAP experiments, and data analysis. P.N. and C.G. designed and provided the GSG-Cx43-P2A-mcherry construct for MMLV production. D.R. and S.E. generated and provided MMLV for *in vitro* and *in vivo* experiments. EC also performed surgeries. T.K., P.K. and C.Z.J. performed and analysed optical mapping. P.K. contributed to the writing of the paper. B.K.F. and W.R. designed the study, co-analysed and discussed data, and wrote the manuscript. All authors have read and approved the final version of the manuscript submitted for publication. B.K.F. and W.R. have equally contributed to this work. All authors agree to be accountable for all aspects of the work. All persons designated as authors qualify for authorship, and all those who qualify for authorship are listed.

## Funding

This work was part of SFB 1425 and funded by the Deutsche Forschungsgemeinschaft (DFG, German Research Foundation; grant #422 681 845) (CZJ, PK, BKF and WR). Optical mapping experiments were funded by the Deutsche Forschungsgemeinschaft (DFG, German Research Foundation; grant #502 822 458 to CZJ).

## Acknowledgements

We thank the members of the Institute of Physiology I (Bonn Germany); namely, S. Grünberg for excellent technical assistance, S. Rieck for technical input on virus/MNP complex formation and M. Funken for support in FRAP analysis.

## Keywords

cardiac gene therapy, connexin 43, magnetic nanoparticles, Moloney murine leukaemia virus, myocardial infarction, ventricular arrhythmias

## Supporting information

Additional supporting information can be found online in the Supporting Information section at the end of the HTML view of the article. Supporting information files available:

**Peer Review History**

