## [Peer Review History · The Journal of Physiology]

Efficient *in vivo* targeting of the myocardial scar using Moloney Murine Leukemia Virus complexed with nanoparticles

Wilhelm Roell, Bernd Fleischmann, Timo Mohr, Miriam Schiffer, Deepak Ramanujam, Esther Carls, Pia Niemann, Caroline Geisen, Stefan Engelhardt, Peter Kohl, Callum Michael Zgierski-Johnston, and Thomas Kok

DOI: 10.1113/JP288020

Corresponding author(s): Wilhelm Roell (wroell@uni-bonn.de)

Review Timeline:

Submission Date:	31-Oct-2024
Editorial Decision:	30-Dec-2024
Revision Received:	15-Mar-2025
Editorial Decision:	07-May-2025
Revision Received:	02-Jun-2025
Accepted:	11-Jun-2025

Senior Editor: Bjorn Knollmann

Reviewing Editor: T Alexander Quinn

Transaction Report:

Dear Dr Roell,

Re: JP-RP-2024-288020 "Efficient in vivo targeting of the myocardial scar using MMLV complexed with nanoparticles" by Wilhelm Roell, Bernd Fleischmann, Timo Mohr, Miriam Schiffer, Deepak Ramanujam, Esther Carls, Pia Niemann, Caroline Geisen, and Stefan Engelhardt

Thank you for submitting your manuscript to The Journal of Physiology. It has been assessed by a Reviewing Editor and by 2 expert referees and we are pleased to tell you that it is potentially acceptable for publication following satisfactory major revision.

Please address all the points raised and incorporate all requested revisions or explain in your Response to Referees why a change has not been made. We hope you will find the comments helpful and that you will be able to return your revised manuscript within 2 months. If you require longer than this, please contact journal staff: jp@physoc.org. Please note that this letter does not constitute a guarantee for acceptance of your revised manuscript.

REVISION CHECKLIST:

We look forward to receiving your revised submission.

Yours sincerely,

Bjorn Knollmann
Senior Editor
The Journal of Physiology

REQUIRED ITEMS FOR REVISION

- Author photo and profile. First or joint first authors are asked to provide a short biography (no more than 100 words for one author or 150 words in total for joint first authors) and a portrait photograph. These should be uploaded and clearly labelled together in a Word document with the revised version of the manuscript. See Information for Authors for further details.

- The contact information for the person responsible for 'Research Governance' at your institution needs to be provided. This includes their name and an institutional email address. Please ensure the contact is not an author on this paper and provide an alternate contact if necessary, or confirm in the submission form that the author whose email was provided has sole responsibility for research governance. This is the person who is responsible for regulations, principles and standards of good practice in research carried out at the institution, for instance the ethical treatment of animals, the keeping of proper experimental records or the reporting of results.

- You must start the Methods section with a paragraph headed Ethical approval (https://jp.msubmit.net/cgi-bin/main.plex?form_type=display_requirements#methods).

Research must comply with The Journal's policies regarding animal experiments (<https://physoc.onlinelibrary.wiley.com/hub/animal-experiments>) and adherence to these policies must be stated in the manuscript.

Authors should confirm in their Methods section that their experiments were carried out according to the guidelines laid down by their institution's animal welfare committee, including an ethics approval reference number. The Methods section must contain a statement about access to food, water and housing, details of the anaesthetic regime: anaesthetic used, dose and route of administration, and method of killing the experimental animals.

- Please upload separate high-quality figure files via the submission form.

- You must upload original, uncropped western blot/gel images (including controls) if they are not included in the manuscript. This is to confirm that no inappropriate, unethical or misleading image manipulation has occurred. These should be uploaded as 'Supporting information for review process only'. Please label/highlight the original gels so that we can clearly see which sections/lanes have been used in the manuscript figures. For more information, see: <https://physoc.onlinelibrary.wiley.com/hub/journal-policies#imagmanip>.

- Please ensure that the Article File you upload is a Word file.

- Papers must comply with the Statistics Policy: https://jp.msubmit.net/cgi-bin/main.plex?form_type=display_requirements#statistics.

In summary:

- If n {less than or equal to} 30, all data points must be plotted in the figure in a way that reveals their range and distribution. A bar graph with data points overlaid, a box and whisker plot or a violin plot (preferably with data points included) are acceptable formats.
- If $n > 30$, then the entire raw dataset must be made available either as supporting information, or hosted on a not-for-profit repository, e.g. FigShare, with access details provided in the manuscript.
- 'n' clearly defined (e.g. x cells from y slices in z animals) in the Methods. Authors should be mindful of pseudoreplication.
- All relevant 'n' values must be clearly stated in the main text, figures and tables.
- The most appropriate summary statistic (e.g. mean or median and standard deviation) must be used. Standard Error of the Mean (SEM) alone is not permitted.
- Exact p values must be stated. Authors must not use 'greater than' or 'less than'. Exact p values must be stated to three significant figures even when 'no statistical significance' is claimed.

- Please include an Abstract Figure file, as well as the Figure Legend text within the main article file. The Abstract Figure is a piece of artwork designed to give readers an immediate understanding of the research and should summarise the main conclusions. If possible, the image should be easily 'readable' from left to right or top to bottom. It should show the physiological relevance of the manuscript so readers can assess the importance and content of its findings. Abstract Figures should not merely recapitulate other figures in the manuscript. Please try to keep the diagram as simple as possible and without superfluous information that may distract from the main conclusion(s). Abstract Figures must be provided by authors no later than the revised manuscript stage and should be uploaded as a separate file during online submission labelled as File Type 'Abstract Figure'. Please also ensure that you include the figure legend in the main article file. All Abstract Figures should be created using BioRender. Authors should use The Journal's premium BioRender account to export high-resolution images. Details on how to use and access the premium account are included as part of this email.

- Please include a full title page as part of your main article (Word) file, which should contain the following: title, authors, affiliations, corresponding author name and contact details, keywords, and running title.

EDITOR COMMENTS

Reviewing Editor:

Your paper has been reviewed by two experts in the field. They felt the approach was innovative and that the study has potential for influence in the field. However, several substantial concerns were raised, which must be addressed for further consideration. As described in the Comments for the Author, this includes issues related to: (i) the need for optimisation for larger or more complex infarcts; (ii) the potential for off-target effects or immune responses; (iii) the lack of long-term data on gene expression stability and potential side effects; (iv) uncertainty about whether Cx43 overexpression could lead to adverse effects like arrhythmias; (v) whether the lower overall transduction rate (relative to the rate in the central scar) can fully explain the observed reduction in ventricular arrhythmias; (vi) a lack of direct assessment of the functional impact on electrophysiological properties and heart function; (vii) small sample size for some experiments, which limits their reliability. If the authors feel they can address the reviewers' concerns, please submit a revised manuscript, along with a point-by-point response to the reviewers' comments.

Senior Editor:

I concur with the reviewing editor's recommendation

REFEREE COMMENTS

Referee #1:

This study explores a gene therapy approach for treating complications following myocardial infarction (MI). MI impairs heart function and causes potentially life-threatening ventricular tachycardias (VT). The study aimed to develop new methods to target scar tissue following MI by modifying cardiac fibroblasts (FB).

The study reports testing a strategy using Moloney Murine Leukemia Virus (MMLV) combined with magnetic nanoparticles (MNP), along with magnetic steering, to enhance the transduction rate of FB. In vitro, this approach successfully transduced approximately 80% of FB. When applied to the forming scar three days after MI-when FB proliferation peaks-the method efficiently transduced the resident fibroblasts.

The study also reports findings using this technique to overexpress Connexin 43 (Cx43), a gap junction protein, in scar tissue. This resulted in the formation of functional gap junctions in vitro and a significant reduction (50%) in the incidence of post-MI VT in vivo. This demonstrates that MMLV/MNP complexes, combined with magnetic steering, are an effective means for targeting and modifying cardiac fibroblasts, potentially improving the functional properties of the cardiac scar, and reducing the risk of VT after MI. The strengths of the study include a highly innovative approach. The use of Moloney Murine Leukemia Virus (MMLV) combined with magnetic nanoparticles (MNP) and magnetic steering to transduce cardiac fibroblasts (FB) is a novel strategy. This approach potentially addresses a critical challenge in gene therapy for heart diseases, particularly post-MI, where targeting the fibroblasts in the forming scar is difficult. This method achieved a high transduction rate (~80%) of cardiac fibroblasts in vitro, suggesting that this technique has strong potential for in vivo applications. The therapy targeted FB three days post-MI, at the peak of fibroblast proliferation is a potentially optimal timepoint for effective treatments directed at post-MI complications. The study demonstrated that overexpressing Connexin 43 (Cx43) in the fibroblasts led to the formation of functional gap junctions and significantly reduced the incidence of post-MI VT by 50%. These findings provide compelling evidence supporting the therapeutic potential of the method in improving heart function and preventing arrhythmias. Finally, the study included in vivo electrophysiological testing, which is critical for translating the results to potential clinical applications, demonstrating the practical applicability of the approach.

The following concerns were identified.

While magnetic steering combined with MMLV/MNP complexes appears efficient in small-scale experiments, scaling up this method for clinical use (e.g., targeting larger or more complex infarcts) may require further optimization and investigation. The study shows efficient transduction of cardiac fibroblasts, but it would be useful to explore whether this method can selectively target fibroblasts without affecting other cell types in the heart, as this could minimize potential side effects.

The study does not evaluate the long-term effects of MMLV/MNP-based gene therapy. For gene therapy approaches, long-term outcomes (e.g., gene expression stability, immune response, or potential tumorigenesis) are crucial to assess before clinical adoption.

The use of viral vectors (like MMLV) for gene delivery can raise concerns about potential immune reactions or off-target effects. The study doesn't fully address these potential risks, which are important considerations in clinical gene therapy.

While overexpressing Cx43 in fibroblasts can form functional gap junctions, it is unclear if overexpression of Cx43 might lead to adverse effects such as arrhythmias or inappropriate electrical coupling (e.g., in post-MI cardiac tissue). The potential for abnormal gap junction formation and functional disturbances should be evaluated, especially in a cardiac setting.

Although the transduction rate in the central regions of the scar area is 19.1%, the overall transduction rate across the entire injection site is only 10.8%. This is relatively low, especially considering that cardiac fibroblasts are difficult to target, and suggests that the method may not yet be efficient enough to explain the changes in the incidence of arrhythmias.

Although the studies show an increase in Cx43 expression, there is no direct assessment of the functional impact of this gene transfer on the electrophysiological properties or contractile function of the infarcted heart. Since the goal of Cx43 overexpression is to enhance gap junction function and potentially improve heart tissue repair, it would be essential to measure functional outcomes such as action potential propagation, gap junction formation, or improvement in heart function.

The sample size of $n = 3$ for some experiments, such as in the imaging and immunostaining assessments, is relatively small. Larger sample sizes would improve the reliability and generalizability of the results, especially in preclinical models where variability can be significant.

Referee #2:

Impact on the area of research:

A very good demonstration of the ability of Cx transfection of fibroblasts within a scar to alter the pro-arrhythmic status of the heart.

Insight into physiological mechanisms in this field:

This study is directed towards methods to moderate the electrocardiography consequences effects of a cardiac scar. The clear effects of increasing the Cx expressed in fibroblasts within the scar reinforces the contention that this approach may be a potential method for the treatment of intractable arrhythmias.

Originality of the research:

The research is a natural extension of the activities of this group over the past few years, but there are novel methods and careful studies on the selectivity and time course of the approach. These describe methods to enhance transduction and estimates of the degree of transfection within the scar.

Study design and robustness of the experimental data:

Good careful study with detailed quantification.

Validity of the conclusions:

The conclusions that Cx over-expression in scar-based fibroblasts alters the arrhythmogenicity of a myocardial scar is well supported by the data. The effectiveness of the approach which involves the use of ferrous particles and focussed magnetic fields to enhance transduction of the scar within the myocardium is well presented.

Criticism:

The study uses sophisticated methods to introduce Cx43 into fibroblasts, but simple methods to assess the pro-arrhythmic effects. The incidence of VF is a good measure of anti-arrhythmic status of the heart, but some questions remain about the mechanism.

(1) The dominant frequency of the VT would may give some insight into what the Cx expression has done to modify the VT circuit in the hearts that retain the VT post transfection. Can the authors please provide that information.

(2) Can the authors supply more information on the nature of the transmural lesion, i.e. the extent of remnant myocardium within the scar. It is difficult to envisage how a patch of fibroblasts within the centre of a lesion can alter the excitability of the myocardium that surrounds the lesion unless there is sufficient remnant myocardium to support some propagation. This information may be available from the histological sections of the scar already available to the authors. What fraction of the scar is occupied by surviving myocardium?

(3) A minor point but there are several instances of the use of the word "infarct" when referring to the lesion produced by cryoprobe. While I accept the technical reasons for using the method and also the idea that the scar is similar to that which is a consequence of an infarction, the more accurate description would be "scar". The lesion results in death of myocardium as a consequence of the cryo-treatment and not the loss of a coronary circulation. Please consider replacing the word "infarct" when describing the scar.

END OF COMMENTS

Universitätsklinikum Bonn, Venusberg-Campus 1, 53127 Bonn

To
Prof. Bjorn Knollman
Senior Editor

The Journal of Physiology
Hodgkin Huxley House
30 Farringdon Lane
London EC1R 3AW, UK

Klinik und Poliklinik
für Herzchirurgie

Univ.-Prof. Dr. med.
Wilhelm Röhl
Oberarzt

Dear Prof. Knollmann, dear Editors,

We would like to thank the reviewers and editors for their positive and constructive comments on our work entitled **“Efficient *in vivo* targeting of the myocardial scar using Moloney Murine Leukemia Virus complexed with nanoparticles”**(JP-RP-2024-288020). We were delighted that the reviewers and editors appreciated the novelty and relevance of improved targeting of the cardiac scar *in vivo*.

We have addressed all the main points mentioned by the reviewing editor and the two reviewers. Below, please find our point-by-point rebuttal. Briefly, we have clarified in the revised manuscript the motivation and limitation of cryoinjury in comparison to other lesion types in mice, and we have addressed the issues related to off-target effects and immune response (new Fig. S3). As requested, we are providing long-term data on gene expression and the anti-VT effect (new Fig. 7; new Fig. 8; new Fig. S6 C,D), as well as explaining the reasons for the apparent lack of adverse effects from the Cx43 overexpression approach. We also include voltage mapping experiments (new Fig. S6 A,B) and discuss the mechanisms underlying Cx43 re-expression in the scar area, the observed reduction of VT incidence *in vivo*, and provide larger sample sizes where possible.

Given the extensive revisions, which include new experiments and analyses we have added Drs. Callum M. Zgierski-Johnston, Thomas Kok, and Peter Kohl from the University of Freiburg as new co-authors for voltage mapping, and named Dr. M. Schiffer as an equally contributing first author.

Tel: 0228 287-14398
0151-18895582
Wilhelm.Roell@ukbonn.de
wroell@uni-bonn.de

Oberarztsekretariat
Fallanmeldung
Anforderung Arztberichte
Tel: 0228 287-14194
Tel: 0228 287-14037
Fax: 0228 287-11634

Universitätsklinikum Bonn
Venusberg-Campus 1
Gebäude 22
53127 Bonn

Ihr Weg zu uns
auf dem UKB-Gelände:

DB6VBC

Vorstand: Univ.-Prof. Dr. Dr.h.c. mult. Wolfgang Holzgreve, MBA, **Vorstandsvorsitzender** und Ärztlicher Direktor • Clemens Platzkoster, Kaufmännischer Direktor und Stellv. Vorstandsvorsitzender • Univ.-Prof. Dr. Bernd Weber, Dekan der Med. Fakultät • Univ.-Prof. Dr. Jörg C. Kalf, Stellv. Ärztlicher Direktor • Alexander Pröbstl, Vorstand Pflege und Patientenservice • **Aufsichtsratsvorsitzender:** Univ.-Prof. Dr. Heinz Reichmann

We hope that our extensively revised manuscript is considered suitable for publication in the *Journal of Physiology* and are looking forward to hearing from you in the near future

Yours sincerely

Wilhelm Röhl, MD

Bernd K. Fleischmann, MD

Referee #1:

This study explores a gene therapy approach for treating complications following myocardial infarction (MI). MI impairs heart function and causes potentially life-threatening ventricular tachycardias (VT). The study aimed to develop new methods to target scar tissue following MI by modifying cardiac fibroblasts (FB).

The study reports testing a strategy using Moloney Murine Leukemia Virus (MMLV) combined with magnetic nanoparticles (MNP), along with magnetic steering, to enhance the transduction rate of FB. In vitro, this approach successfully transduced approximately 80% of FB. When applied to the forming scar three days after MI-when FB proliferation peaks-the method efficiently transduced the resident fibroblasts.

The study also reports findings using this technique to overexpress Connexin 43 (Cx43), a gap junction protein, in scar tissue. This resulted in the formation of functional gap junctions in vitro and a significant reduction (50%) in the incidence of post-MI VT in vivo. This demonstrates that MMLV/MNP complexes, combined with magnetic steering, are an effective means for targeting and modifying cardiac fibroblasts, potentially improving the functional properties of the cardiac scar, and reducing the risk of VT after MI. The strengths of the study include a highly innovative approach. The use of Moloney Murine Leukemia Virus (MMLV) combined with magnetic nanoparticles (MNP) and magnetic steering to transduce cardiac fibroblasts (FB) is a novel strategy. This approach potentially addresses a critical challenge in gene therapy for heart diseases, particularly post-MI, where targeting the fibroblasts in the forming scar is difficult. This method achieved a high transduction rate (~80%) of cardiac fibroblasts in vitro, suggesting that this technique has strong potential for in vivo applications. The therapy targeted FB three days post-MI, at the peak of fibroblast proliferation is a potentially optimal timepoint for effective treatments directed at post-MI complications. The study demonstrated that overexpressing Connexin 43 (Cx43) in the fibroblasts led to the formation of functional gap junctions and significantly reduced the incidence of post-MI VT by 50%. These findings provide compelling evidence supporting the therapeutic potential of the method in improving heart function and preventing arrhythmias. Finally, the study included in vivo electrophysiological testing, which is critical for translating the results to potential clinical applications, demonstrating the practical applicability of the approach.

Answer: We thank the reviewer for his/her very positive assessment of our work and for the constructive comments to further improve the quality of our work.

The following concerns were identified.

- 1) While magnetic steering combined with MMLV/MNP complexes appears efficient in small-scale experiments, scaling up this method for clinical use (e.g., targeting larger or more complex infarcts) may require further optimization and investigation. The study shows efficient transduction of cardiac fibroblasts, but it would be useful to explore whether this method can selectively target fibroblasts without affecting other cell types in the heart, as this could minimize potential side effects.

Answer: We thank the reviewer for bringing up this important point. We have chosen the MML virus because it is known to selectively transduce proliferating cells, particularly cardiac (myo)fibroblasts (Ramanujam D. et al., Molecular Therapy 2016). As a result, cardiomyocytes, which are terminally differentiated in adult mammals, are thought to be not transduced and hence unaffected by this virus. Importantly, we have carefully searched for mCherry signals in heart sections and did not observe any mCherry⁺ cells outside of the scar area (Fig. S2C). This indicates that neither cardiomyocytes were transduced, nor did transduced myofibroblasts migrate from the scar into the remote myocardium.

While MMLV can also transduce endothelial cells, as we have shown in vitro (Fig. S4), following gene therapy in vivo we only detected a few PECAM⁺ endothelial cells in the scar area (Figs. 4D,E). There are two possible explanations: Proliferating endothelial cells were transduced at the time of injection but did not survive until the analysis at two- and eight weeks post-injection, and/or there were no proliferating endothelial cells at the time of injection, which would preclude transduction with MMLV. As previously reported, myofibroblasts in the developing cardiac scar proliferate strongly between 2 and 4 days after myocardial infarction (Fu et al., JCI 2018). We have chosen this time point for gene therapy to facilitate the selective transduction of myofibroblasts in the cardiac lesion. Furthermore, since we did not detect a prominent immunological response (new Fig. S3), we noted no major adverse/or off-target effects from the local intracardiac injections. While it cannot be entirely ruled out that MMLV may enter the bloodstream via the coronary vessels, the combined use of MNP and the magnet greatly enhances the retention of the virus-MNP complexes.

We agree that scalability is an issue, and therefore, the proposed approach needs to be transferred to a large animal model that exhibits more complex and larger myocardial infarcts. Due to these differences, the MNP type, as well as the magnet, will need to be adapted. We have added this point to the revised version of the Discussion.

- 2) The study does not evaluate the long-term effects of MMLV/MNP-based gene therapy. For gene therapy approaches, long-term outcomes (e.g., gene expression stability, immune response, or potential tumorigenesis) are crucial to assess before clinical adoption.

Answer: We thank the reviewer for highlighting this important point. As requested, we have performed and analysed mice long-term (8 weeks) following MMLV injection (new Figs. 7, 8, S3, S6 C,D). In these mouse hearts, we still observed the transduction of cells in the scar area. The number of cells (approximately 8,000) was, as expected, lower compared to the 2-week time point, as the scar thins out over time, leading to an overall reduction of cellularity by approximately 50%, as quantified by our group recently. Importantly, MMLV-Cx43 treated mice continue to show significant protection against VT at this later time point compared to both cryo-injured and MMLV-mCherry treated cryo-injured control mice (VT incidence: CI-Cx43: 50%, CI: 83%, CI-mCherry: 85%, new Fig. 8). Furthermore, we have not observed adverse effects such as an increased immune response, as the number of CD45⁺ cells in the scar was comparable to the 2-week time point (new Fig. S3), and no tumors were observed. However, as explained in the Discussion section, other retroviruses apart from MMLV should be used for humans given their potential to insert into protooncogenic sites.

- 3) The use of viral vectors (like MMLV) for gene delivery can raise concerns about potential immune reactions or off-target effects. The study doesn't fully address these potential risks, which are important considerations in clinical gene therapy.

Answer: As the injection of a virus triggers an immune response, mice were treated with a relatively mild immunosuppressive regimen for 10 days post-injection (for details, see the Methods section). There was no increased immune response in the scars of animals at 2- and 8 weeks following injection based on CD45 assessment (new Fig S3). The Engelhardt group (Ramanujam D. et al., Molecular Therapy 2016) has shown that MMLV injection into the pericardial cavity in neonatal mice did not result in the transduction of cells in other organs (kidney, liver, lung, spleen). Furthermore, we have not observed any adverse effects, and due to the intracardiac, magnet-driven virus-MNP injection systemic and/or off-target effects are not anticipated.

- 4) While overexpressing Cx43 in fibroblasts can form functional gap junctions, it is unclear if overexpression of Cx43 might lead to adverse effects such as arrhythmias or inappropriate electrical coupling (e.g., in post-MI cardiac tissue). The potential for abnormal gap junction formation and functional disturbances should be evaluated, especially in a cardiac setting.

Answer: As mentioned above, we have not observed any adverse effects from using the MMLV-Cx43 treatment, as no cardiomyocytes are transduced by MMLV. It is primarily the formation of Cx43 hemichannels in cardiomyocytes that is believed to potentially cause problems (Leybaert L. et al., JCI 2023). In contrast, our approach of overexpressing Cx43 in cardiac scar cells demonstrates strong anti-VT activity in vivo. This is further underscored by careful analysis of the electrophysiological in vivo data over time (2 vs. 8 weeks, new Fig. S6 C-F). While VT events per animal increased significantly over time in CI- as well as in CI-mCherry control animals, the number of VT events in CI-Cx43 animals remained stably low over time (Fig. S6C). In addition, the analysis of VT duration (Fig. S6D) and VT frequency (Fig. S6E) showed no difference among the various groups of mice or between short- and long-term cohorts. Importantly, the post-operative survival rate is also very similar between CI-mCherry and CI-Cx43 mice, ruling out major Cx43 dependent adverse effects. We have included this information in the revised Discussion section of the manuscript.

- 5) Although the transduction rate in the central regions of the scar area is 19.1%, the overall transduction rate across the entire injection site is only 10.8%. This is relatively low, especially considering that cardiac fibroblasts are difficult to target, and suggests that the method may not yet be efficient enough to explain the changes in the incidence of arrhythmias.

Answer: The reviewer raises an interesting point.

First, the transduction rate of MMLV in the scar is much higher than in previous studies that used different viruses. For example, when using lentivirus, only approximately 1.1-2.8% of scar cells were transduced (Roell et al Sci.Rep. 2018). We agree with the reviewer that, ideally, higher transduction numbers would be preferable. However, we present a strategy in which gene targeting of the cardiac scar becomes feasible. Improved viruses, MNP, and magnet design, along with testing different delivery routes in large animals, could further increase homogeneity and overall transduction numbers. We have included this important point in the revised Discussion of the manuscript.

Second, the observation that only a low number of grafted cardiomyocytes or myofibroblasts overexpressing Cx43 in the murine scar are needed to strongly reduce VT-incidence in vivo is a consistent feature reported in our earlier work (Roell et al., Nature 2007, Roell et al Sci.Rep. 2018). We

have observed that this leads to enhanced conduction velocity in the border zone and scar of the lesion and reduces wave breaks and re-entry-VT.

- 6) Although the studies show an increase in Cx43 expression, there is no direct assessment of the functional impact of this gene transfer on the electrophysiological properties or contractile function of the infarcted heart. Since the goal of Cx43 overexpression is to enhance gap junction function and potentially improve heart tissue repair, it would be essential to measure functional outcomes such as action potential propagation, gap junction formation, or improvement in heart function.

Answer: We thank the reviewer for raising this point: The primary goal and novelty of this work is to establish effective in vivo targeting of the cardiac scar. To achieve this, we have utilized a newly designed MMLV construct complexed with MNP and magnetic steering. In order to demonstrate that this approach has a functional impact, we employed the Cx43 overexpression strategy, which our group has previously shown to protect against VT in vivo. We demonstrate through molecular, cell biological, and functional (FRAP) assays that using this novel virus construct, Cx43 is overexpressed in (myo)fibroblasts in vitro and in the scar area of the mouse heart, and functional gap junctions are formed. Additionally, we also provide echocardiography measurements indicating that left ventricular function is, as expected, unaltered at short- and long-term following virus injection (Fig.6, new Fig.8). We have, as requested by the reviewer, performed optical mapping experiments in Langendorff-perfused hearts at 2 weeks post-treatment (new Fig S6A, B). These experiments clearly demonstrate a reduction in the heterogeneity of conduction between remote and scarred tissue with Cx43 overexpression, reducing the likelihood of re-entrant arrhythmias. This is consistent with our earlier work, where conduction in the border zone and scar is increased, going towards normal levels (Roell et al., Nature 2007, Roell et al. Sci.Rep. 2018). However, due to technical limitations such as dye loading, the small size of the mouse heart, and the scar area, we have been unable to determine whether enhanced heterocellular coupling and/or electrotonic events are responsible for the observed anti-VT effects (see above). As explained in the Discussion section, experiments in a larger animal model are warranted to further elucidate the cellular mechanisms in greater detail, which is clearly beyond the scope of this manuscript.

- 7) The sample size of $n = 3$ for some experiments, such as in the imaging and immunostaining assessments, is relatively small. Larger sample sizes would improve the reliability and generalizability of the results, especially in preclinical models where variability can be significant.

Answer: We have increased, where feasible, sample sizes (new Fig. 1C,D,F,G,H; new Fig. 2B,D; new Fig. 3B,C,D; new Fig. S4B,C).

Referee #2:

Impact on the area of research:

A very good demonstration of the ability of Cx transfection of fibroblasts within a scar to alter the pro-arrhythmic status of the heart.

Insight into physiological mechanisms in this field:

This study is directed towards methods to moderate the electrocardiography consequences effects of a cardiac scar. The clear effects of increasing the Cx expressed in fibroblasts within the scar reinforces the contention that this approach may be a potential method for the treatment of intractable arrhythmias.

Originality of the research:

The research is a natural extension of the activities of this group over the past few years, but there are novel methods and careful studies on the selectivity and time course of the approach. These describe methods to enhance transduction and estimates of the degree of transfection within the scar.

Study design and robustness of the experimental data:

Good careful study with detailed quantification.

Validity of the conclusions:

The conclusions that Cx over-expression in scar-based fibroblasts alters the arrhythmogenicity of a myocardial scar is well supported by the data. The effectiveness of the approach which involves the use of ferrous particles and focussed magnetic fields to enhance transduction of the scar within the myocardium is well presented.

Criticism:

The study uses sophisticated methods to introduce Cx43 into fibroblasts, but simple methods to assess the pro-arrhythmic effects. The incidence of VF is a good measure of anti-arrhythmic status if the heart, but some questions remain about the mechanism.

Answer: We thank the reviewer for his/her appreciation of our work and the very constructive and helpful comments.

- 1) The dominant frequency of the VT would may give some insight into what the Cx expression has done to modify the VT circuit in the hearts that retain the VT post transfection. Can the authors please provide that information.

Answer: We have analysed the R-R interval during VT and during sinus rhythm immediately thereafter (new Fig. S6 E,F) and found no difference in mice that developed VT between groups (~1200 bpm during VT, 450-500 bpm basal heart rate at sinus rhythm) suggesting that overexpression of Cx43 does not influence the frequency of VT. Moreover, we have also performed optical mapping in Langendorff-perfused mouse hearts at 2 weeks post-treatment, illustrating that conduction is more homogeneous in hearts overexpressing Cx43, which will render re-entrant arrhythmias less likely to occur (see new Fig. S6 A,B).

Can the authors supply more information on the nature of the transmural lesion, i.e. the extent of remnant myocardium within the scar. It is difficult to envisage how a patch of fibroblasts within the centre of a lesion can alter the excitability of the myocardium that surrounds the lesion unless there is sufficient remnant myocardium to support some propagation. This information may be available from

the histological sections of the scar already available to the authors. What fraction of the scar is occupied by surviving myocardium?

Answer: There are no surviving cardiomyocytes in the scarring cryolesion. In fact, at 7 days after cryolesion, no α -actinin⁺ cells are detected within the scar (Fig. S2 C). Thus, this type of lesion differs from left coronary ligation models, in which islands of surviving cardiomyocytes can be found in the scar area.

The propagation of electrical signals from the border zone to the more central areas following MMLV-Cx43 transduction/Cx43 overexpression has not yet been established. This remains a highly relevant question for future studies (see also below), which may only be feasible in a large animal model. Importantly, Cx43 overexpressing myofibroblasts are found not only in the center but also, though in lower numbers, in the periphery of the lesion, potentially enabling electrotonic coupling. In our previous work, we aimed to fully understand the cellular mechanisms underlying the anti-VT effect (Roell et al., Nature 2007; Roell et al., SciRep. 2018). We consistently found that a relatively low number of cells overexpressing Cx43 in the cardiac scar can significantly reduce the incidence of VT in vivo. This is likely due to enhanced conduction velocities in the border zone and scar area, as well as a reduction in wave breaks and re-entry VT, which is consistent with the observed reduction in the heterogeneity of conduction between remote and scarred tissue in the optical mapping experiments shown in new Fig. S6 A,B. However, due to the small lesion size and heart dimensions in mice, the high heart rate, and difficulties with dye loading, we have been unable to determine whether heterocellular coupling, electrotonic events, and/or other cellular mechanisms underlie this strong anti-VT protection. As explained in the Discussion section, experiments in a larger animal model are warranted to further elucidate the cellular mechanisms in greater detail, which is clearly beyond the scope of this manuscript.

- 2) A minor point but there are several instances of the use of the word "infarct" when referring to the lesion produced by cryoprobe. While I accept the technical reasons for using the method and also the idea that the scar is similar to that which is a consequence of an infarction, the more accurate description would be "scar". The lesion results in death of myocardium as a consequence of the cryo-treatment and not the loss of a coronary circulation. Please consider replacing the word "infarct" when describing the scar.

Answer: We would like to thank the reviewer for pointing this out and have replaced "infarct" with either lesion or scar in the revised version of the manuscript. Furthermore, it is stated in the text and in the figures that a cryoinjury (CI) was used to generate the cardiac lesions.

Dear Dr Roell,

Re: JP-RP-2025-288020R1 "**Efficient *in vivo* targeting of the myocardial scar using Moloney Murine Leukemia Virus complexed with nanoparticles**" by Wilhelm Roell, Bernd Fleischmann, Timo Mohr, Miriam Schiffer, Deepak Ramanujam, Esther Carls, Pia Niemann, Caroline Geisen, Stefan Engelhardt, Peter Kohl, Callum Michael Zgierski-Johnston, and Thomas Kok

Thank you for submitting your revised Research Article to The Journal of Physiology. It has been assessed by the original Reviewing Editor and Referees and has been well received. Some final revisions have been requested.

REVISION CHECKLIST:

We look forward to receiving your revised submission.

Yours sincerely,

Bjorn Knollmann
Senior Editor
The Journal of Physiology

REQUIRED ITEMS FOR REVISION

- You must start the Methods section with a paragraph headed Ethical approval (https://jp.msubmit.net/cgi-bin/main.plex?form_type=display_requirements#methods).

Research must comply with The Journal's policies regarding animal experiments (<https://physoc.onlinelibrary.wiley.com/hub/animal-experiments>) and adherence to these policies must be stated in the manuscript.

Authors should confirm in their Methods section that their experiments were carried out according to the guidelines laid down by their institution's animal welfare committee, including an ethics approval reference number. The Methods section must contain a statement about access to food, water and housing, details of the anaesthetic regime: anaesthetic used, dose and route of administration, and method of killing the experimental animals.

PLEASE NOTE - Your paper contains Supporting Information of a type that we no longer publish, including supplementary tables and figures. Any information essential to an understanding of the paper must be included as part of the main manuscript and figures. The only Supporting Information that we publish are video and audio, 3D structures, program codes and large data files. Your revised paper will be returned to you if it does not adhere to our Supporting Information Guidelines.

EDITOR COMMENTS

Reviewing Editor:

Thank you for your extensive and responsive revision; you have satisfied all of the reviewers' concerns.

Senior Editor:

The revised MS is improved and now potentially acceptable after all supplemental figures have been incorporated into the main MS. Please revise the MS accordingly

"Material such as figures, tables, text (e.g. expanded/detailed methods or results), equations, and other material, if essential for the complete understanding of the manuscript and can fit on a printed page or pages, must be incorporated into the manuscript itself as part of the text or as standard figures or tables and not supplied as supporting information."

REFEREE COMMENTS

Referee #1:

Concerns addressed

Referee #2:

I have read the revised ms and the response of the reviewers and acknowledge that the authors have addressed my concerns fully with significant edits and the addition of new data.

I have no further critical comments on the manuscript.

END OF COMMENTS

Klinik und Poliklinik für Herzchirurgie, Venusberg Campus 1, 53127 Bonn

Klinik und Poliklinik
für Herzchirurgie

Bonn, 29. Mai 2025

**Univ.- Prof. Dr. med.
Wilhelm Röhl
Oberarzt**

Dear Prof. Knollmann, dear Editors,

We would like to thank the reviewers and editors for their positive evaluation of our revised manuscript entitled “Efficient in vivo targeting of the myocardial scar using Moloney Murine Leukemia Virus complexed with nanoparticles” (JP-RP-2024-288020R1), and for potentially accepting it for publication pending the implementation of some formal issues.

In our revised manuscript, we have addressed the remaining formatting issues. As requested, and in accordance with the Journal’s guidelines, we included all figures and tables in the main manuscript.

We hope our revised manuscript is considered suitable for publication in the *Journal of Physiology*, and we look forward to hearing from you in the very near future.

Yours sincerely,

Dr. W. Röhl, MD

Dr. B. Fleischmann, MD

Tel: 0228 287-14398
Fax: 0228 287-14195
wroell@uni-bonn.de

Anschrift:
Universitätsklinikum Bonn
Klinik und Poliklinik für Herzchirurgie
Venusberg Campus 1
53127 Bonn

Dear Professor Roell,

Re: JP-RP-2025-288020R2 "**Efficient *in vivo* targeting of the myocardial scar using Moloney Murine Leukemia Virus complexed with nanoparticles**" by Wilhelm Roell, Bernd Fleischmann, Timo Mohr, Miriam Schiffer, Deepak Ramanujam, Esther Carls, Pia Niemann, Caroline Geisen, Stefan Engelhardt, Peter Kohl, Callum Michael Zgierski-Johnston, and Thomas Kok

We are pleased to tell you that your paper has been accepted for publication in The Journal of Physiology.

Yours sincerely,

Bjorn Knollmann
Senior Editor
The Journal of Physiology

If you would like to receive our 'Research Roundup', a monthly newsletter highlighting the cutting-edge research published in The Physiological Society's family of journals (The Journal of Physiology, Experimental Physiology, Physiological Reports, The Journal of Nutritional Physiology and The Journal of Precision Medicine: Health and Disease), please click this link, fill in your name and email address and select 'Research Roundup':
<https://www.physoc.org/journals-and-media/membernews>

- You can help your research get the attention it deserves! Check out Wiley's free Promotion Guide for best-practice recommendations for promoting your work at: www.wileyauthors.com/eeo/guide. You can learn more about Wiley Editing Services which offers professional video, design, and writing services to create shareable video abstracts, infographics, conference posters, lay summaries, and research news stories for your research at: www.wileyauthors.com/eeo/promotion.

EDITOR COMMENTS

Reviewing Editor:

The additional requirements have been fulfilled.

Senior Editor:

The MS is now acceptable. Thank you for your excellent contribution to the Journal!